



# Transforming school students' aspirations into destinations through extended interaction with cutting-edge research: 'Physics Research in School Environments'

Martin O. Archer[1], Jennifer DeWitt[2,3], and Charlotte Thorley[4]

[1]School of Physics and Astronomy, Queen Mary University of London, London, UK
[2]Institute of Education, University College London, London, UK
[3]Independent Research and Evaluation Consultant, UK
[4]Public Engagement and Involvement Consultant, UK

**Correspondence:** Martin O. Archer
(martin@martinarcher.co.uk)

**Abstract.** We introduce a scalable framework for protracted research-based engagement with schools called 'Physics Research in School Environments' (PRiSE) which has transformed cutting-edge space science, astronomy, and particle physics into accessible 6-month independent research projects for schools. The programme's theory of change presents how PRiSE aims to impact on a diverse range of 14–18 year-old students, supporting and enhancing their physics aspirations, as well as influencing

teachers' practice and their school environments to potentially enable wider impacts. We explore the considerations made in developing the programme to help enact these theorised changes, in particular detailing the structure, support, and resources offered by active researchers as part of PRiSE. Through feedback from participating students and teachers, we assess the provision within this framework. This illustrates that the model appears to provide highly positive experiences that are otherwise not accessible to schools and that the extraordinary level of support offered is deemed necessary with all elements appear-

ing equally important. Researchers and public engagement professionals seem receptive to the PRiSE framework of schools engagement and it has started to spread to other institutions.

## 1   Introduction

Research, policy, and practice all agree that participation in Science, Technology, Engineering and Mathematics (STEM) needs to be increased and widened (e.g. Campaign for Science and Engineering, 2014), with these issues being particularly

acute for the subject of physics (e.g. Murphy and Whitelegg, 2006; IOP, 2014). Physics as a field has become within society strongly aligned with intelligence/cleverness, masculinity and whiteness, all of which can dissuade school students (even those highly-enthusiastic about the subject) from pursuing it further and thereby showing inequitable effects on those from under-represented backgrounds (L. Archer et al., 2020a). Some of these issues arise from practices in school-level physics education. Debarring and gatekeeping of physics based on attainment (disproportionately so compared to other subjects) simply feeds

the alignment of physics with cleverness and can make even high-attaining students' confidence in the subject precarious. Teachers and the school environment often (even unconsciously) reinforce stereotypes about physics and physicists that are





patterned by biases. Curriculum practices in physics often teach oversimplifications at younger ages which are later completely reconceptualised without being presented as refinements to a model, making students perceive the simpler versions as "lies". Furthermore the general deferment of "interesting" physics in the curriculum produces a disconnect between "school physics"

and "real physics", i.e. the cutting-edge research undertaken by professional physicists, making continued participation in physics education something of a "test of endurance". These concerns are further reflected in results from national surveys. While 20% of 16–18 year-old physics school students in the UK aspire towards a physics degree and 80% aspire towards STEM more broadly (Wellcome Trust, 2017), only 9.7% and 59.3% actually go on to study either physics or STEM respectively (McWhinnie, 2012). These constitute odds ratios for aspirations vs. destinations of 2.3 and 2.7, both of which are considerable.

All of these issues raised cultivate and contribute to reproducing inequitable, and low overall, patterns of participation within physics (L. Archer et al., 2020a).

Davenport et al. (2020) suggest that for STEM subjects in general an intervention approach that sustains and supports science identity is most appropriate for students in late secondary/high school education in the context of their educational journey. However, in the case of physics specifically, L. Archer et al. (2020a) comment that existing interventions based on simply

enthusing, inspiring and informing students about physics will not significantly change uptake or diversity in post-compulsory physics. While they advocate for widespread changes in science education policy and practice, both at school and university levels, they note that if interventions are also used they need to fundamentally address the problematic processes and practices present within both physics teaching and physics as a field generally.

The stark differences between school, university, and professional science practices have long been noted — while research

is one of the main activities of professional scientists, it is quite removed from how science is taught in schools with some arguing science education is not "authentic" in this respect (e.g. Hodson, 1998; Braund and Reiss, 2006). Indeed, Yeoman et al. (2017) report that school students are largely unaware of what research actually is, finding a disconnection between 'research as information gathering' and the 'research question', and in general have little opportunity to set their own research questions within their school environment. Independent research projects, which provide extended opportunities for students

to lead and tackle open-ended scientific investigations, may be one way of incorporating "real science" into schools. These align with established international pedagogical initiatives such as 'inquiry-based science' (e.g. Minner et al., 2010), 'problem-based learning' (e.g. Gallagher et al., 1995) and 'authentic science' (e.g. Braund and Reiss, 2006). A survey of such projects, however, found them to be rare globally and are only sometimes supported by mentors from university/industry (Bennett et al., 2016, 2018). The review found considerable variability in the nature of independent research projects such as their focus,

delivery/provision models, external support, and funding/costs. It was noted that such programmes place demands on time and money beyond standard provisions for all stakeholders, on the skills required by teachers and other adults involved, and on the supporting infrastructure. For successful projects Dunlop et al. (2019) recommend that students should be given the freedom to devise a research question, have ownership over their own data analysis and decision-making, and be given access to experts in their project work.

A subset of independent research projects have been termed 'research in schools', which concern projects for schools that are linked to current academic (STEM) research. While many citizen science projects also aim to help participants (which





can include school students) learn about current science and to experience the scientific research process, these are typically secondary aims with most citizen science projects primarily concerning a single (or small number) of well-defined science questions which will be assisted through developed citizen science protocols (Bonney et al., 2009, 2016; Shah and Martinez,
2016). This contrasts with independent research projects, and thus also 'research in schools', where positively affecting the participants is the primary concern and the projects are necessarily open-ended. Nonetheless, the different approaches can have some overlap and indeed some projects denoted as citizen science, such as the 'curriculum-based' projects described by Bonney et al. (2016), might perhaps be better framed as 'research in schools'. In the UK we are aware of three 'research in schools' programmes with projects in the physical sciences outside of that at Queen Mary University of London (QMUL),
which forms the subject of this paper and is introduced in section 2.

    HiSPARC (High School Project on Astrophysics Research with Cosmics) is a scintillator-photomultiplier cosmic ray detector project originating in the Netherlands which has been adopted by UK universities including Bath, Bristol, Birmingham and Sussex (Colle et al., 2007; HiSPARC, 2018). Many of these universities operate a tiered membership scheme for schools: 'Gold' enables schools to buy their own detector at £5,500; 'Silver' is a detector rental scheme (£300 p.a. plus an installation
fee) with the contract specifying if they do not participate the detector will be collected with an additional fee; and 'Bronze' membership (£200 p.a.) gives schools access to HiSPARC data but not their own detector. Schools signing up for 'Silver' or 'Bronze' membership are contractually obliged to generate funding to upgrade to 'Gold'. While the 'Gold' membership fee covers the costs of the detector and installation, the other memberships are seemingly justified to ensure that schools make a commitment to working with the university (J. Velthuis and M. Pavlidou, personal communication, 2016; National HE STEM
SW, 2012). The HiSPARC website lists 22 UK schools as hosting detectors, likely 'Gold' and 'Silver' members, though it is not stated how many schools have been engaged in total. It is not possible to compare how these schools go about project work and how much support they are given by participating universities, which may vary by institution, as at the time of writing HiSPARC has not published any reviews of their processes or evaluation.

    IRIS (Institute for Research in Schools) is a UK charity formally launched in March 2016 (IRIS, 2020), building on the
previous CERN@School project conceived in 2007 (Whyntie et al., 2016; Parker et al., 2019). While IRIS's projects cover all the sciences, current physics projects include the aforementioned CERN@School, Higgs Hunters (Barr et al., 2018), LUCID (Furnell et al., 2019; Hatfield et al., 2019), and Webb Cosmic Mining (in preparation for the James Webb Space Telescope). They have rapidly expanded across the UK since formation, having worked in some capacity with over 230 schools as of 2020. Publications have provided technical details of their projects and case studies of some students' successes within them,
including a few examples of resulting peer-reviewed scientific work, however the exact provision/delivery model implemented and precisely how project work is supported is not fully explored in the available literature. IRIS aims to develop 'teacher scientists', teachers that identify as both science teachers and research-active scientists (Rushton and Reiss, 2019), which suggests a teacher-driven model. While some researchers/academics have designed or consulted on some IRIS projects, they appear in general to have little involvement supporting students or teachers (O. Moore, personal communication, 2020). With
a recent change of staff at IRIS in late 2019 has come a reformulation of how they classify their projects. 'Seed' projects are for new schools, are the most straightforward, and receive the most support from IRIS though it is not clear in what form that





takes. 'Sprout' projects are more advanced seeing students carrying out more complex activity to assist scientists with their research questions, though how this collaboration operates is not specified. 'Grow' projects are where students have proposed their own research questions, either independently or using IRIS resources, with IRIS merely providing advice in producing

posters, talks, or papers as well as opportunities to present.

ORBYTS (Original Research By Young Twinkle Scientists), based at University College London, was piloted from January 2016 and is nominally based around the Twinkle mission, though has expanded into other research areas since (Sousa-Silva et al., 2018; ORBYTS, 2019). A select group of students (with an imposed limit of 4–6) from each school undertakes fortnightly meetings with early career researchers (either PhD students or post docs) throughout their project aiming to achieve, where

possible, publishable scientific results (McKemmish et al., 2017; Chubb et al., 2018; Holdship et al., 2019). Teachers, while present, are not typically actively involved in these sessions and students tend to do little independent work outside of the sessions (W. Dunn, personal communication, 2018). The projects' content changes each year to align with the researchers' current focus, with them typically working with only one school per year each. PhD researchers are paid for their (preparation, travel, and session) time with funds from independent schools, who pay not only for their school but in enabling an additional

school from a lower socio-economic background to take part.

It is clear that there is currently a lack of published details in general on both the considerations towards and practicalities of provision within the emerging area of 'research in schools' projects. This paper therefore explores these aspects, with the aim of informing others' schools engagement practice, applied specifically to the 'research in schools' programme of QMUL's School of Physics & Astronomy. This programme was piloted between 2014–2016, as detailed in M.O. Archer (2017), and is

now known as 'Physics Research in School Environments' (PRiSE, 2020). Section 2 introduces the aims of the programme through a theory of change. Based on these aims, the considerations in developing and evolving PRiSE's framework since the pilot are discussed, with the structure and support (both through interventions and produced resources) as well as details of current projects all being explored in detail. The framework is evaluated in section 4 using feedback from participating students and teachers to assess their experience through PRiSE and whether the provision offered is sufficient. We also briefly discuss

how the PRiSE approach appears to have been received by the university sector in section 5. Two companion articles to this paper explore the diversity of schools and equity of the programme (M.O. Archer, 2020), as well as investigating the impact of PRiSE on participating students, teachers, and schools (M.O. Archer and DeWitt, 2020).

## 2   PRiSE framework

The 'Physics Research in School Environments' (PRiSE) programme at QMUL grew out of a need communicated by teachers

for more in-depth schools engagement activities featuring bespoke content and repeated interventions (M.O. Archer, 2017). Out of these discussions it was deemed that the cutting-edge physics research being undertaken was a unique aspect to universities that other types of providers of schools engagement or informal science education (e.g. museums, freelance science communicators) are unable to offer, hence should be capitalised upon. This moved away from much of the schools engagement by university physics departments at the time, where the majority of activities were largely highly tied to the curriculum (e.g.





Galliano, 2015; SEPnet, 2017). While curriculum-based engagement has been identified as a category of informal science education by Lloyd et al. (2012), being one way teachers look to deliver and enhance the required educational material, they also recognised the broader area of voluntary non-curriculum linked activity which can encompass other aims and benefits to students. Furthermore, physics researchers are largely unmotivated in delivering curriculum content as part of their schools engagement work, whereas aspects relating to their research and role as a researcher are much more valued by them (Thorley,

2016). There is a spectrum of how research-embedded schools engagement activities can be — from little-to-no research content at all; to one-way engagements such as a talk that may reference the research; research-inspired two-way activities such as in a workshop; and up to fully involving school students in the actual research process. PRiSE was developed to be as close to the latter level as realistically possible and the move to more protracted and two-way research-embedded schools engagement, while fairly nascent within the sector at the time, was well-founded based on the stakeholders' motivations.

The purpose of this section is to explore in detail our thinking behind the development of PRiSE so that others who may wish to adopt this model are able to better understand fully the framework and motivations behind its numerous aspects. We explain the aims of the programme through a theory of change, frame the ethos underlying the programme and its provision, and discuss in depth the considerations made in PRiSE's structure and support mechanisms.

## 2.1 Aims

There is no "magic bullet" to increasing physics (or more broadly STEM) uptake and diversity at higher education — multiple different approaches are needed with each addressing different stages of young people's educational journey as well as their key influencers and wider learning ecology in relevant ways (e.g. Davenport et al., 2020). Furthermore, research has shown that young people's aspirations are incredibly difficult to influence (L. Archer et al., 2013, 2014) with standard one-off (or even short-series of) intervention(s) showing no real changes, highlighting the need for more extended and in-depth programmes for

significant lasting impact (M.O. Archer et al., Under Review, and references therein).

Given this complexity, we detail the aims of the PRiSE programme through a theory of change (Sullivan and Stewart, 2006). These are designed to rationalise the outcomes and impacts of an initiative by outlining causal links. The process of creating a theory of change works backwards, starting at the intended ultimate impact and mapping the intermediate outcomes (both short-, medium-, and long-term) that are thought to be required to enable that goal. The resulting outcomes pathway (which may

require iterating several times) should be accompanied by the rationale for why specific connections exist between different outcomes in the theory narrative along with any underlying assumptions. Figure 1 displays the theory of change for PRiSE, which covers participating students (blue) as well as their parents/carers (yellow) and their teachers and school environment (both red).

The intended impact of PRiSE is to contribute towards the increased uptake and diversity of physics at higher education.

By serving students near the end of their school-based educational journey, somewhat necessitated by the content and style of open-ended 'research in schools' projects, the programme acts to support students' existing identity with science in general and enhance, or at least maintain, physics aspirations to help transform these into degree subject destinations — a known issue at this stage. Students' interest or enjoyment in the subject as well as its perceived usefulness in a career are key factors





**Figure 1.** The Theory of Change for PRiSE.





affecting degree choices (DeWitt et al., 2019), with students (particularly girls) often thinking physics is less useful or relevant

(Murphy and Whitelegg, 2006). Additionally, the stereotypes and school-based practices associated with physics make many, even highly-able and interested students, at this age conclude it is 'not for me' (L. Archer et al., 2020a). PRiSE attempts to be a factor in addressing all of these factors in some way. By interacting first-hand with "real physics" through the projects and working with active researchers, students (especially those from under-represented groups) should feel included and have their interest in physics enhanced or at least sustained. By experiencing success at 'being' a scientist and meeting similar students

from other schools, it is hoped their confidence will be boosted leading to a feeling that physics is indeed for 'people like me' (Davenport et al., 2020). Furthermore, through working in new ways students should develop numerous transferable skills (Bennett et al., 2018) which might help them recognise the usefulness of the subject (Soh et al., 2010).

Teachers are much stronger influences on students' aspirations than university staff/students could ever be (L. Archer et al., 2013, 2020b). Experience from physics outreach officers (e.g. through discussions via the South East Physics Network's Out-

reach and Public Engagement and Ogden Trust's Outreach Officer programmes) have shown that most teachers are more interested in activities for their students from universities rather than continuing professional development opportunities, which they may seek elsewhere. Therefore, opportunities for teachers' development are integrated within the programme rather than being a separate offering to schools. While the number of students working on PRiSE may be relatively small, by influencing teachers through our sustained programme, the aim is that the impacts of PRiSE can be felt much wider. Indeed, our hope is

to affect the environments within the diverse range of schools we work with on the programme so that they are places that are able to support and nurture the science capital (L. Archer et al., 2013, 2020b) of all their students, thereby also contributing to our goal of increased uptake and diversity of physics (Moote et al., 2019, 2020). IOP (2014) recommends to help achieve such an environment that schools should raise the overall profile of science in school, endeavour to build long-term relationships between pupils and role models, and ensure all teachers are aware of the influence they can have on children's future

careers. We aim to further all these points by PRiSE providing more collaborative working opportunities between teachers and students, exciting success stories that teachers and students can share across their schools, and the gateway to building longer-term relationships between schools and the university.

Finally, another major influence on young people's aspirations are family, particularly parents or carers (e.g. Clemence et al., 2013). Parental engagement is notoriously difficult within school-based programmes (M.O. Archer et al., Under Review), so

we simply aim to include parents/carers to celebrate in students' project work at the end of the programme. It is hoped that by witnessing their child's successes and development through physics, they will be more positive, and thus supportive, towards physics aspirations going forward, reinforcing the impacts of the programme.

Whether the PRiSE framework discussed in this paper is successful at enacting these theorised impacts is evaluated in a companion paper (M.O. Archer and DeWitt, 2020). We do, however, stress that this theory of change does not exist in isolation

and other theories of change which focus on different stages of and aspects to a young person's learning ecology are required to improve the overall issue of uptake and diversity in physics (cf. Davenport et al., 2020; M.O. Archer et al., Under Review).





| | Academic Year | 2014/2015 | 2015/2016 | 2016/2017 | 2017/2018 | 2018/2019 | 2019/2020 |
|---|---|---|---|---|---|---|---|
| | Schools | 1 | 6 | 18 | 29 | 27 | 33 |
| Number per year | Students | 20 | 115 | 163 | 310 | 311 | 407 |
| | Teachers | 1 | 7 | 25 | 29 | 31 | 38 |
| | Schools | 1 | 6 | 20 | 39 | 50 | 67 |
| Unique cumulative total | Students | 20 | 135 | 298 | 608 | 919 | 1326 |
| | Teachers | 1 | 7 | 28 | 44 | 63 | 88 |

**Table 1.** Number of schools, students and teachers involved in PRiSE by academic year as the programme has grown.

## 2.2 Reach

Since PRiSE's pilot between 2014–2016, the programme has been grown carefully but substantially. This was done to increase the number of schools we are able to work with while still maintaining the provision offered. Table 1 indicates the number
of schools, students and teachers that have been involved by academic year, demonstrating PRiSE now serves around 30 schools per year having reached a total of 67 schools and over 1,300 students with the direct involvement of 88 teachers as of 2020. A full analysis of the types of schools involved is given in M.O. Archer (2020). Programmes of repeat-interventions with schools will necessarily have a smaller reach than various one-off events and only well-developed programmes will have built the capacity to expand while still ensuring quality and success. However, protracted schools engagement programmes
are still fairly embryonic within university STEM outreach / public engagement, as noted in the recent landscape review of M.O. Archer et al. (Under Review). For example of the other physical science 'research in schools' projects/programmes in the UK: University of Oxford researchers have interacted directly with only 14 students from 5 schools through their Higgs Hunters IRIS project which commenced in 2016 (O. Moore, personal communication, 2020); ORBYTS (2019) reports collaborating with 17 schools since 2016; and HiSPARC (2018), adopted in the UK since 2012, lists 22 schools on their website shared
amongst four UK universities. Therefore the reach of PRiSE by a single university department is considerable, for the depth of interaction, compared to the rest of the sector and has been achieved due to our engagement framework, which aims to find a balance between the (necessarily competing) reach and significance of the interactions.

## 2.3 Approach

PRiSE takes the 'research in schools' approach to schools engagement, whereby students are given the opportunity to lead
and tackle open-ended scientific investigations in areas of current research. Therefore, the PRiSE projects were developed to transform current scientific research methods, making them accessible and pertinent to school students so that they could experience, explore, and undertake open-ended scientific research themselves.

One might think it is feasible that students' work on PRiSE projects contribute to novel research. However, we stress that the primary focus of PRiSE is (unlike typical citizen science) on the participants rather than the research. Our position is that
it is rather unreasonable to expect investigations that are motivated by school students themselves (an established element of





good practice in independent research projects, e.g. Dunlop et al., 2019) to be able to make meaningful contributions to the physics research as a matter of course. We note that in some exceptional cases PRiSE students' work has arrived at promising preliminary results, though these have required significant follow-up work by professional researchers to transform the results into publishable research. Students and teachers have been credited as co-authors in resulting publications in such cases (e.g.

M.O. Archer et al., 2018). These outcomes, however, should be considered as rare benefits rather than the archetype. This is generally true also of other 'research in school' schemes (B. Parker, personal communication, 2017), with perhaps the exception of ORBYTS due to its explicit aim on this necessitating more researcher-driven projects. Given these practicalities, arguably the main advantage for PRiSE researchers is through evolving their engagement practice beyond typical one-off approaches (cf. M.O. Archer et al., Under Review) and reflecting on their research through both transforming their methods to be accessible

to school students and subsequently interacting with young people via this research project. This two-way interaction between the research/researchers and schools with the aim of mutual benefit means that PRiSE is a programme of public engagement with research (National Coordinating Centre for Public Engagement, 2020). Nonetheless, the aims of positively influencing students' aspirations/destinations with regards to physics or STEM, as well as developing teachers' practice, also heavily overlap with the typical goals of outreach programmes with schools (e.g. Thorley, 2016).

Since the programme intends to influence school students and teachers a number of ethical considerations have been taken into account, following the BERA (2018) guidance for educational research, with regard to safeguarding and to ensure that no harm results. Firstly, to ensure equality of access to the programme we do not charge schools to be involved (cf. Harrison and Shallcross, 2010; Jardine-Wright, 2012) and try to provide them with all the physical resources they need for their project, thereby removing potential barriers to entry for less resourced schools. Our targeting takes into account several school-level

metrics to ensure diversity and we aim for the programme to be equitable to all (see M.O. Archer, 2020) with all schools being offered the same interventions/opportunities, taking into account and being flexible to their specific needs where necessary. We work with as many schools as we have capacity to do so each year and do not withold interventions from any students for the purpose of having control groups.

A key part with regard to safeguarding is the involvement of teachers at all stages. They have helped shape the design of the

programme, inform how we update it each year, and serve as our liaison to schools and the students involved. It is the teachers that decide who projects are offered to within their school, with us simply advising that the projects should be suitable for all A-Level (16–18 year old) students as well as high-ability GCSE (14–16 year-old) students (further contextual information on the UK education system is given in Appendix A). These recommendations were made based on the basic background knowledge required to meaningfully engage with the research. Invariably teachers choose to involve older age groups, with $79 \pm 1\%$ of

PRiSE students being aged 16–18 (and so far only one student below our recommended ages has been involved, being 13–14). The projects are optional and presented as an opportunity that students can take advantage of which will be supported by their teacher and the university, therefore students are not pressured into being involved. Students and schools can drop out at any point within the programme with no penalty. We allow teachers to determine how best to integrate the projects within their school, though provide advice on this. We also aim, through our resources and communications, to equip teachers to manage





the day-to-day aspects of the projects without overly burdening them — their role is chiefly one of encouraging their students to persist, providing what advice they can, and then communicating with the university.

It was recognised that teachers in general likely will not have the skills or experience in research to manage projects without expert assistance (Shah and Martinez, 2016; Bennett et al., 2016, 2018). Therefore, PRiSE was designed to be supported by active researchers equipped with the necessary expertise to draw upon in offering bespoke, tailored guidance to the students

and teachers. Well-defined roles within the university have been established for each of the PRiSE projects to provide this support:

– **Outreach Officer:** This role manages the entire programme including university-school relationships, communications, intervention/event co-ordination, programme finances and evaluation. At QMUL this role has been performed by the first-author.

– **Project Lead:** This is a member of staff with considerable research experience in the topic area who acts as a visible figurehead for the project to schools. Ideally this person also leads the project's (iterative) development and plays an active role its delivery throughout, however, that has unfortunately not been the case for all PRiSE projects at QMUL.

– **Researcher:** This role can be undertaken by any researcher with experience in the topic area, providing advice and guidance to students and teachers throughout the programme. While some project leads also take on the researcher role,

in some cases this is either delegated to or shared with an early career researcher (such as a post-doc or PhD student).

Details of these key personnel for the current PRiSE projects are given in section 2.5. While the programme management falls within the scope of the funded outreach officer post, a role which many university departments now employ (e.g. Ogden Trust, 2020), whether to pay early career researchers or assign workload allocations to academics would realistically be down to the policy/strategy of the department or institution. In the case of Queen Mary, we have opted to only offer pay to PhD

student researchers (sourced from the department's outreach budget) to try to increase uptake of engagement, whereas workload allocation in outreach / public engagement for academics is dedicated only to roles aimed at embedding engagement throughout the department, with delivery of engagement being considered an expectation of the role of an academic which may be used as criteria for promotions.

## 2.4 Structure

PRiSE runs from the start of the UK academic year to just before the spring/Easter break, which teachers had informed us during the pilot stage is manageable and largely fits around exams / other activities for most (but not necessarily all) schools (M.O. Archer, 2017). The structure has evolved naturally from the pilot to that shown in Table 2, which emerged from 2017 onwards.

### 2.4.1 Intervention and activity stages

Here we detail the different intervention and activity stages that form the structure of PRiSE:





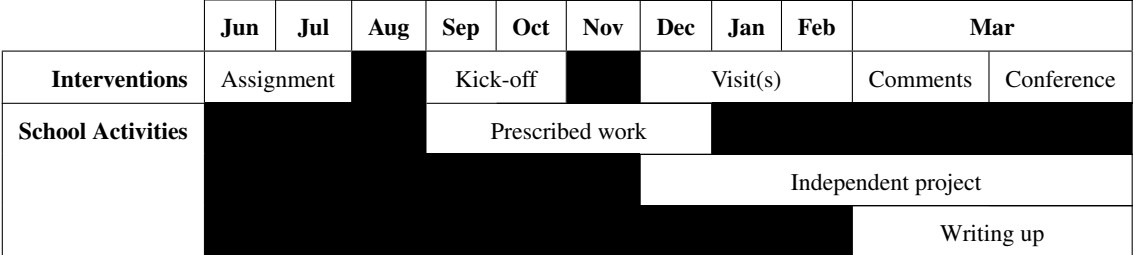

| | Jun | Jul | Aug | Sep | Oct | Nov | Dec | Jan | Feb | Mar | |
|---|---|---|---|---|---|---|---|---|---|---|---|
| **Interventions** | Assignment | | | Kick-off | | | | Visit(s) | | Comments | Conference |
| **School Activities** | | | | Prescribed work | | | | | | | |
| | | | | | | | Independent project | | | | |
| | | | | | | | | | Writing up | | |

**Table 2.** Structure of PRiSE programme.

– **Assignment:** We advertise the opportunity to school teachers for the following academic year largely using existing teacher networks such as the Institute of Physics' Stimulating Physics Network (Hartley, 2011) and the Ogden Trust School Partnerships (Ogden Trust, 2020). Using these networks not only allow us access to schools from lower socio-economic areas given the networks' focus but also act somewhat like a word-of-mouth recommendation — we have found these networks to be more successful at attracting new schools to the programme than our existing schools events mailing lists. Teachers fill out an online form providing school and contact details, project preferences, and the estimated number of students who will be involved. Previously participating schools have to reapply each year. Once applications are in we assess the capacity of the programme (taking into account data about the schools, see M.O. Archer, 2020) and inform teachers before the summer break whether their school has been allocated a project or not. Most schools are assigned only one project, which both helps with logistics and makes it easier for teachers to manage, and where possible we take into account their stated preferences though this is not always possible given the researchers' workloads. At the start of the academic year we send posters/flyers to teachers to help attract attention to the project within their school.

– **Kick-off:** These events are hosted either in-school, sometimes within a normal lesson or at lunchtime/after-school depending on the teacher, or as an evening event on (university) campus. Projects led by an academic member of staff are typically on campus due to constraints on their time, though in some cases where schools could not attend the event we have repeated it at their school. Kick-off events start with a 20–30 minute introductory talk by the project lead concerning the underlying physics and research topic, leading up to an overview of what the project is about, which is presented to students as an opportunity that they can take advantage of if they wish. The outreach officer then discusses the differences between learning styles in the research project compared to their regular classroom experience, how the project will work, the support available, and how to go about obtaining this. The event ends with a hands-on workshop for at least 20 minutes usually run by the outreach officer and facilitated by researchers (though not always the project lead). This workshop typically forms either the early part of or a lead into the initial stage of the project work. Experts are on hand to assist with any questions or initial troubles, with the aim of getting the students and teachers to a place where they can continue this work without too much extra help for the next month or so. Students and teachers are given all the resources (see later) they need to begin/continue their project work from this point on.





– **Prescribed work:** Students work in research groups of typically five people and they are advised to try and work on the project on average for 1–2 hours a week. The bulk of this is done outside of regular physics lessons, though some schools integrate the projects within their timetabled 'science clubs' or required extra-curricular blocks, whereas other teachers arrange a regular slot for students to work on the projects or leave it up to the students to arrange (though this latter approach often proves unsuccessful). Given that independent research in STEM is probably unfamiliar to the students, rather than expecting them to be able to come up with their own avenues of investigation in an unfamiliar research topic straight away, we instead give them an initial prescribed stage of research which is detailed in their student guide. This involves following a set of instructions to undertake an experiment/activity designed to cover most aspects of an investigation and to build their confidence in the project topic. Students are still required to problem solve throughout these stages and we purposely do not provide them with all the answers to prompt this.

– **Visit:** These school visits by researchers are often administered through a rolling Doodle (http://www.doodle.com) poll where teachers can sign up to a session given the researcher's availability. Schools taking advantage of this stage typically receive only one visit, though if further demand is communicated we try to accommodate 1–2 additional ones. The visits typically last around an hour and occur around the stage where groups have finished the prescribed activities and are thinking about or are in the early stages of undertaking their independent research. They are very much student-driven meetings, where the researcher asks the groups of students to show what they have done, probes their understanding of this, puts their work into the context of current research, provides answers to any questions the students have, and gives advice on what the direction and next steps with their specific project ideas might entail while bearing in mind what methods/results may be achievable within the timeframe of the project. Only active researchers have the necessary expertise to draw upon in offering such bespoke, tailored guidance to students and teachers working on projects in their research area. With one project (ATLAS) and for a few schools on other projects it has not been possible to have in-person visits for logistical reasons, however, similar interactions were done via specifically arranged Skype calls to the schools in these cases.

– **Webinars:** One project (MUSICS) has experimented with additional support to schools through monthly drop-in webinars between November–February, providing further opportunities for students and teachers to ask questions of the researcher and get advice on how to progress with their project work in a similar manner to the visits. This was first trialled in 2018/19 through a Google Hangout simultaneously streamed on YouTube, however, this option was later discontinued so a solution using a Skype group call also broadcast to YouTube (via the NDI® feature and using Open Broadcast Software, https://obsproject.com/) was implemented in 2019/20. The YouTube streams are unlisted so that only project students with a link can access them, making the webinars a safe space for them to discuss the project. While almost all students and teachers preferred to simply join the YouTube stream and contribute via its live chat facility, the rationale behind incorporating the Hangout/Skype option was so that participants could directly talk to the researcher and/or show their work. In 2018/19 the webinars were organised in a somewhat ad hoc manner and due to technical limitations the only way of communicating the links to join was via an email immediately before the webinar.





With the move to Skype we were able to create a stable hyperlink to join the group as well as being able to embed the YouTube events in advance on a password-protected webpage, both of which allow for easier access to the webinars. In terms of organisation, at the beginning of the 2019/20 academic year we sent out an online form asking teachers to iden-tify when might be the best times for webinars. While the response rate for this was low (only four), we used this to set a regular monthly schedule (in this case the first Monday of each month at 4–5pm) which was communicated to teachers

far in advance. All these changes considerably increased the uptake of webinars: 10 out of 14 schools participated in at least one webinar in 2019/20 compared to only 2 out of 14 the previous year.

– **Independent project:** Following the prescribed work, groups are encouraged to set their own research questions and undertake different projects in the topic area, continuing in a similar way to with the prescribed work except now inde-pendently motivated. This enables every group across all schools to explore something different so that students gain a

sense of independence and ownership in their own work and not feel that they are doing exactly the same as everyone else. In visits and webinars the question has been raised by students whether there is a risk that they investigate the same thing as another group at a different school, though given the broad scope of most of the PRiSE projects so far this has rarely occurred. Potential research questions are suggested in the guides provided and students' ideas are discussed during visits and/or webinars.

– **Writing up:** Near the end of the project students produce either a scientific poster or talk to be presented at our annual conference. Guidance on how to approach these is provided online as well as during visits and webinars.

– **Comments:** Students are offered the opportunity to receive comments on their draft slides or posters near the end of the project, in much the same way as researchers would receive comments on their work from collaborators. These are currently given by the outreach officer, though in general this would depend on their background/experience and could

instead be done by the relevant researcher role(s). Teachers (or students directly) email their work to the outreach officer and receive annotated versions back the week before the final deadline, allowing the students at least a few days to implement any changes.

– **Conference:** Students present the results of their projects at a special conference, 'Cosmic Con', held at Queen Mary. This is attended by researchers as well as the students' teachers, peers, and family. The evening is primarily based

around oral and poster presentations by the students. Food and drink are provided during the poster session and we also put on various physics demonstrations. At the end of the evening all student groups are congratulated and given a thank you letter, with a select number of groups highlighted by researcher judges also receiving prizes in the form of various science gadgets (some prize winners have also had the opportunity to present their research at a national student conference hosted by the Royal Society). As of 2019 we limited the number of talk slots available to four in total, both

for time and so each topic can be covered. Schools are only able to solicit one talk (where desired) by providing a title and abstract in advance (early March), with the decisions of who will present being made that same week. There are currently no limits on the number of posters a school can enter into the conference.





Photos depicting some of these stages are displayed in Figure 2. In addition to these interventions, ad hoc support is also provided via email where required. We explain at the kick-off meetings that when students get stuck at any point (which they

invariably will do due to the nature of research) they should try to first tackle this themself, before discussing in their groups, and then raising with their teacher. In general, teachers act as the primary contact to students offering encouragement and any support or advice they can. If students' questions go beyond what their teacher can answer and is not covered by the teacher guides we provide, the teachers should get in touch through the outreach officer. Some teachers, however, instruct their students to email directly. Not all schools require this option and we have not yet been overloaded with additional questions. Only in

a few cases has the outreach officer not been able to directly answer the question, subsequently passing it on to a relevant researcher to answer, though in general this would depend on the background and experience of the outreach officer.

All of the stages of the programme and the processes involved are communicated to teachers via email to pass on to their students. The outreach officer typically sends updates and reminders to all teachers involved fortnightly throughout the programme, which attempts to tackle teachers' generally low levels of response to a single email (Sousa-Silva et al., 2018, also

reported teachers' generally poor communication through ORBYTS). In addition to email communications, we also set up a password-protected teacher area on our website in 2019 which always contains the latest information on the programme. It appears from website analytics that teachers have used this to some extent (there were 76 unique page views amongst the 38 teachers involved that year), though we are unsure which teachers these were and how often they visited the page throughout the programme. For any on campus events, schools travel to Queen Mary using London's extensive public transport network

and we are unable to offer schools compensation for travel expenses due to limited funding. Outside of London or other well-connected cities this travel may be a greater barrier to participation than in our case so may need further consideration.

### 2.4.2 Resources

To enable the students to take part in PRiSE, the students and teachers are also provided with numerous resources. While specific equipment, data and software pertaining to individual projects are detailed later, here we discuss more common types

of resources across the different projects.

Each project has a student guide, presented in the style of an academic paper. Printed copies of these are given out at the kick-off event and electronic version can also be found on the project's page on our website. These serve as an introduction, providing enough information for students to start working on their project and be something they can refer back to throughout. However, the guide is not intended to be exhaustive (that would be impossible given the open-ended nature of the projects) and students

are made aware that we expect them to read additional materials as they progress. Generally these guides cover the following areas: an introduction to the research field, background physics/theory, an explanation of the equipment/data, discussion of analysis techniques, details of the initial prescribed activity, suggested research questions / methods for independent research, and links to other sources of information. Throughout the guides there are exercises and discussion questions for students to consider, designed to help them think more critically about their project work. Teachers are provided with the same guide, but

with extra guidance including answers to the exercises, hints and tips about different methods, common pitfalls that students make etc. and these are distributed to teachers at the kick-off, via email, and also stored on the password-protected teacher



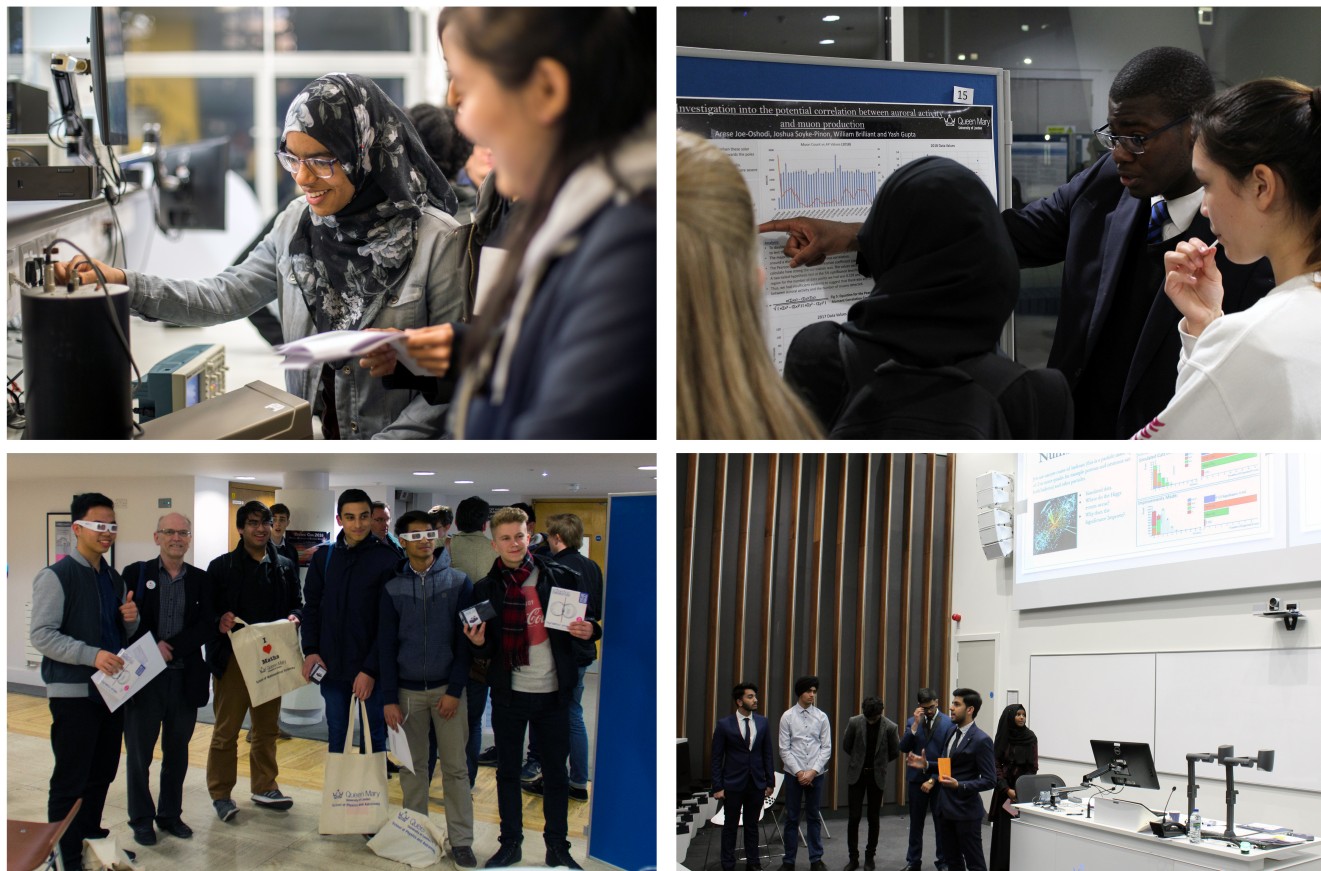

**Figure 2.** Photos of various stages of the PRiSE framework: students participating in an on campus kick-off workshop (top left), students interacting during the poster session at a conference (top right), a group of students display their prizes won at a conference along with their teacher (bottom left), a group present a talk at a conference (bottom right).

webpage. These project guides have been updated every single year based on feedback and professional experience, which is straightforward to do given the chosen style compared to say a more illustrated glossy guide that would require a professional designer each time.

On project webpages we also showcase anonymised examples of good quality talks/posters that previous students have produced as part of their project work. This had been suggested by students and teachers in feedback for a few years as something that would be helpful. Additionally some projects have produced videos which provide further information on the science / research area or demonstrate how to use tools provided for the project.

Finally, general 'how to' style articles have been produced by the outreach officer for students that are applicable to all
projects. The most used of these are the guides on producing scientific talks and posters that we point students and teachers to ahead of the conference. While articles in this section designed for teachers have also been planned, which would highlight





elements of good practice that have emerged from other teachers on how to successfully integrate and nurture project work within their schools outside of the support offered by researchers, we have not had sufficient time or detailed input from teachers to be able to co-create these yet.

## 2.5 Current projects

QMUL's physics research areas concern astronomy (space science, planetary physics, and cosmology), particle physics (the Standard Model and beyond via particle colliders and neutrino observatories), condensed matter physics (e.g. material structure, organic semiconductors, and applications thereof), and theoretical physics (e.g. string theory, and scattering amplitudes). Of these, it was decided to initially base PRiSE around the space and planetary sciences as well as particle physics. These exciting topics are thought to inspire awe in the public due to the "big" questions they address and the senses of scale and wonder beyond our everyday experience (cf. Madsen and West, 2003; IPPOG, 2020). However, exactly how this science is conducted is not often well understood outside of academia, particularly at school-level due to the lack of research methods within current science teaching (e.g. Hodson, 1998; Braund and Reiss, 2006; Yeoman et al., 2017). Currently four projects have been developd for the PRiSE programme at QMUL, which we briefly summarise here indicating key project personnel (referring to the roles mentioned previously).

- **Scintillator Cosmic Ray Experiments into Atmospheric Muons (SCREAM, 2014–2020)** was adapted by the outreach officer from a dissertation project designed for undergraduates using a scintillator – photomultiplier tube muon detector (Coan and Ye, 2016; TeachSpin, 2016) fundamentally similar to those found in current neutrino experiments such as SNO+, where cosmic ray muons serve as an important background source that can be used for calibration (Alves et al., 2015). Students calibrate their borrowed detectors and collect counts of comic ray muons and muon decays. Initially they use this data to perform a measurement of muons' mean lifetime (using both software that comes with the detector and programmes we have created especially) before progressing to a wide variety of potential topics on these cosmic rays such as their angular distributions or dependence on atmospheric/solar conditions. Since detector usage and particle physics are part of most A-Level physics syllabuses, it complements their studies. At present QMUL only has four of these detectors as they are expensive (around £5,000), which limits the number of schools we can work with each year. As this project has been especially popular with teachers when signing up, we have made it open only to schools that have successfully undertaken a different project with us previously, also limiting how long they can borrow a detector to a maximum of 4 years.

  Key personnel: Dr Jeanne Wilson (project lead, 2014–2019), Prof Peter Hobson (project lead, 2019–2020), Dr Martin Archer (researcher, 2015–2020)

- **Magnetospheric Undulations Sonified Incorporating Citizen Scientists (MUSICS, 2015–2020)** was created especially for PRiSE by the project lead. Geostationary satellite data of the "sounds of space", ultra-low frequency fluid plasma waves in Earth's magnetosphere, have been made audible. Students are given this data on preloaded USB flash drives and explore it through the act of listening (we also provide them with earphones). Audacity audio soft-





ware (https://www.audacityteam.org/) is used to analyse any events identified, which can then be logged in a specially created spreadsheet which performs some routine calculations. We stress that students do not have to focus on the space plasma physics aspects, which will largely be completely unfamiliar, but rather just the waves topics that they cover in class both at GCSE and A-Level. While students are given guidance on how to listen to and analyse the waves, we do not prescribed to them exactly what to listen out for as we are instead interested in what they pick out themselves. This

approach has already led to novel and unexpected scientific results on the resonances present in Earth's magnetosphere during the recovery phase of geomagnetic storms (M.O. Archer et al., 2018). Based on these results, an optional 'solar storms campaign' was created for 2019/20, providing more concrete prescribed instructions to build up a dataset of similar events followed by suggestions of unanswered questions about these resonances that students could investigate. While a few schools followed this route initially, they all eventually decided to go their own way with it.

Key personnel: Dr Martin Archer (project lead and researcher, 2015–2020)

- **Planet Hunting with Python (PHwP, 2016–2020)** was initially developed by a post-viva PhD student, Dr Gavin Coleman, through a one-day-per-week buyout over three months (funded by a grant obtained by the outreach officer) and has subsequently been modified by the project lead each year. The project aims to address the UK coding agenda by applying Python computer programming to data from NASA's Kepler (Jenkins et al., 2010) and later TESS (Ricker et al.,

2015) missions, whereby students write programmes to detect exoplanet transits. Transit photometry, where an exoplanet blocks some of the star's light, can be fairly easily understood by school students in terms of geometry and the equations from A-Level physics. The students try to independently implement each step laid out in their guide (period detection, phase folding, model fitting, and parameter estimation) applied to specially selected star systems chosen for their relative simplicity. Example code is given to teachers. While extension activities are suggested, so far very few students have

progressed beyond the prescribed activities within a single year, though some students have returned for a second year making further progress.

Key personnel: Prof Richard Nelson (project lead, 2016–2020, and researcher, 2017–2020), Dr Gavin Coleman (researcher, 2016–2017), Dr Martin Archer (researcher, 2016–2017), Francesco Lovascio (researcher, 2020)

- **ATLAS Open Data (2017–2020)** was adapted by the outreach officer for PRiSE from a public resource produced by

CERN aimed at undergraduates (ATLAS Experiment, 2017). An undergraduate summer student, under the instruction of the outreach officer, produced a guide so that school students could build up to the documentation provided online by CERN. At kick-off workshops students play a loaded dice game, developed by the outreach officer and freely-available online as a resource (PRiSE, 2020), which serves as an analogy for why particle physicists need to use statistical methods and big data in discovering new particles such as the Higgs boson (ATLAS Collaboration, 2012). This leads into the main

activity, using CERN's interactive histograms to see how performing cuts on the data increase/decrease the significance of the desired signal, i.e. the Higgs, compared to the backgrounds. While the CERN guides provide extensions by using their statistical software (ROOT) for more detailed analysis, this has been beyond almost all PRiSE students thus far, with most groups simply investigating the underlying physics behind their chosen cuts to justify them.





Key personnel: Dr Eram Rizvi (project lead, 2017–2019), Dr Seth Zenz (project lead, 2019–2020, and researcher, 2018–
2020), Joe Davies (researcher, 2019–2020)

It is clear that the topics and activities vary considerably, much like research activities across fields of physics. However, the ethos behind all of these projects' design align with the PRiSE programme overall — that of providing students and teachers an accessible way of exploring on their own terms cutting-edge research science topics and methods — and are all delivered / supported using the PRiSE framework. This suggests that a wide range of fields and project ideas might be able to adopt 490   the PRiSE framework. Of the current projects at QMUL, only MUSICS at present has the scope to lead to novel publishable scientific research (which it already has done). The Kepler dataset has largely been mined of the clearest exoplanets, often now requiring advanced machine learning techniques for new discoveries (e.g. Shallue and Vanderburg, 2018) which are currently also being implemented on TESS. The other two projects have limitations based on the equipment (Coan and Ye, 2016; TeachSpin, 2016) and amount of data used (ATLAS Open Data's first release contained only a fifth of the data used 495   in the Higgs boson's discovery, ATLAS Collaboration, 2012, however, more data was released in 2020). While this is not perhaps ideal, we have adopted a pragmatic approach in taking advantage of university opportunities and adapting existing materials where possible, since creating a project from scratch is a significant undertaking far beyond what most academics (unfortunately) have capacity to do (cf. Thorley, 2016). This is further compounded when PRiSE projects are not embedded within their research groups. M.O. Archer (2017) recommended following the PRiSE pilot that 'research in schools' projects' 500   development and delivery should be distributed within each research group sharing the load out amongst academics, post-docs, and PhD students, in turn allowing more schools to participate without overburdening individual researchers. However, this research group buy-in has proven difficult to achieve at Queen Mary and responsibilities have remained largely been falling to only a few people per PRiSE project. Nonetheless, there have been some positive steps in the last year with project leads, along with the outreach officer, being able to convince a few early career researchers to help with delivery, which may indicate 505   the department slowly moving towards a more embedded approach.

## 3   Methods

To determine the perceived value and effectiveness of PRiSE's approach with its key stakeholders, namely participating students and teachers as well as those across the wider university sector, we have maintained regular collection of evaluative data (cf. Rogers, 2014, and references therein) via various surveys which we detail here. This data underpins our understanding of 510   PRiSE in this and other papers, and has been collected securely to protect all participants, in compliance with GDPR and in line with the BERA (2018) guidelines for educational research.

### 3.1   Instruments and participants

We gathered feedback from participating students and teachers via paper questionnaires handed out at our student conferences each year. The only exception to this was in 2020, where online forms were used due to the COVID-19 pandemic causing that 515   year's conference to be postponed. The questionnaire method was chosen so as to gather data from as wide a range of students





|  | 2015 | 2016 | 2017 | 2018 | 2019 | 2020 |
|---|---|---|---|---|---|---|
| **Students** |  | 13/26 (50%) | 21/70 (30%) | 46/92 (50%) | 38/97 (39%) | 35/? |
| **Teachers** | 1/1 (100%) | 6/6 (100%) | 6/11 (55%) | 9/16 (56%) | 6/16 (38%) | 17/? |
| **Schools** | 1/1 (100%) | 6/6 (100%) | 11/ 11 (100%) | 13/15 (87%) | 11/15 (73%) | 19/? |

**Table 3.** Response rates to questionnaires at PRiSE student conferences.

and teachers as possible, respecting the limited time/resources of all involved (both on the school and university sides). For ethics considerations all feedback was anonymous, with students and teachers only indicating their school (pseudonyms are used here to protect anonymity) and which project they were involved with. Students were not asked to provide details of any protected characteristics (such as gender or race) or sensitive information (such as socio-economic background). Both students

and teachers were informed via an ethics statement on the form that the information was being collected for the purpose of evaluating and improving the programme and that they could leave any question they felt uncomfortable answering blank (this functionality was also implemented on the online form for consistency).

The open and closed questions concerning participants' experience of the programme, which varied slightly year-to-year, are given in Appendix B. The questionnaires also included questions regarding impact, which are explored in another paper

(M.O. Archer and DeWitt, 2020). While we attempted to collect responses from all participants in attendance, invariably only a fraction did so yielding results from 153 students and 45 teachers across 37 schools. A breakdown of the number of respondents and their schools per year is given in Table 3, where the number of participants and schools in attendance at our conferences are also indicated (retention within the programme is discussed in M.O. Archer, 2020). We do not have reliable information on how many students, teachers, and schools would have successfully completed the programme in 2020 due to the COVID-19

disruption. Students and teachers did not always answer all of the questions asked, hence we indicate the number of responses for each question considered throughout. There is no indication that the respondents differed in any substantive way from the wider cohorts participating in the programme. While ideally one would also gather feedback from schools which dropped out during the year, a similar formal feedback process has not been viable bar in a few cases where only the teachers responded.

Feedback from the university sector came from a session at the 2019 Interact symposium (M.O. Archer, 2019), where an

anonymous interactive online survey was integrated into the workshop. The survey included both closed and open questions as listed in Appendix C. Attendees were fairly evenly split between UK university researchers and engagement professionals (gauged in-person by attendees raising their hands when asked), with 19 people participating in the survey and only 7 not doing so. Participants were allocated a unique number by the online survey itself, which did not distinguish between researchers and engagement professionals.

**3.2 Analysis**

Both qualitative and quantitative approaches were utilised in data analysis, as the open and closed ended questions present in the questionnaires produced different types of data.





For all quantitative data, standard (i.e. 68%) confidence intervals are presented throughout. For proportions/probabilities these are determined through the Clopper and Pearson (1934) method, a conservative estimate based on the exact expression

for the binomial distribution, and therefore represent the expected variance due to counting statistics only. Several statistical hypothesis tests are used with effect sizes and two-tailed $p$-values being quoted, with the required significance level being $\alpha = 0.05$. In general we opt to use nonparametric tests as these are more conservative and suffer from fewer assumptions (e.g. normality, interval-scaling) than their parametric equivalents such as t-tests (Hollander and Wolfe, 1999; Gibbons and Chakraborti, 2011). The Wilcoxon signed-rank test is used to compare single samples to a hypothetical value, testing whether

differences in the data are symmetric about zero in rank. When comparing unpaired samples a Wilcoxon rank-sum test is used, which tests whether one sample is stochastically greater than the other (often interpreted as a difference in medians). Finally, for proportions we use a binomial test, an exact test based on the binomial distribution of whether a sample proportion is different from a hypothesized value (Howell, 2007). For ease of reference, further details about the quantitative analyses are incorporated into the relevant sections of the findings.

Qualitative data were analysed using thematic analysis (Braun and Clarke, 2006). Instead of using a priori codes, the themes were allowed to emerge naturally from the data using a grounded theory approach (Robson, 2011; Silverman, 2010) as follows:

1. Familiarisation: Responses are read and initial thoughts noted.

2. Induction: Initial codes are generated based on review of the data.

3. Thematic Review: Codes are used to generate themes and identify associated data.

4. Application: Codes are reviewed through application to the full data set.

5. Analysis: Thematic overview of the data is confirmed, with examples chosen from the data to highlight the themes.

## 4    Feedback from participants

In this section we use the feedback from participating students and teachers to evaluate the provision offered within the PRiSE framework, specifically assessing their experience and the level of support offered.

### 4.1    Experience

Firstly from 2016 onwards we asked both students ($n = 150$) and teachers ($n = 42$) "Have you been happy with the research project overall?" giving options on a 5-point Likert scale, which we coded to the values 1–5. This scale and the results are displayed in Figure 3, revealing that $91 \pm 3\%$ of students and $95 \pm 5\%$ of teachers rated their experience as positive (scores of 4–5) with only three students giving a negative reaction (scores of 2). Teachers tended to rank this question somewhat

higher (their mean score was $4.50 \pm 0.09$, where uncertainties refer to the standard error in the mean) than students (mean of $4.17 \pm 0.05$), with $p = 0.002$ in a Wilcoxon rank-sum test. The PHwP project scored slightly higher (average of $4.59 \pm 0.11$, $p = 8 \times 10^{-4}$) than the overall results with students, whereas ATLAS scored slightly lower with both students ($3.92 \pm 0.09$,





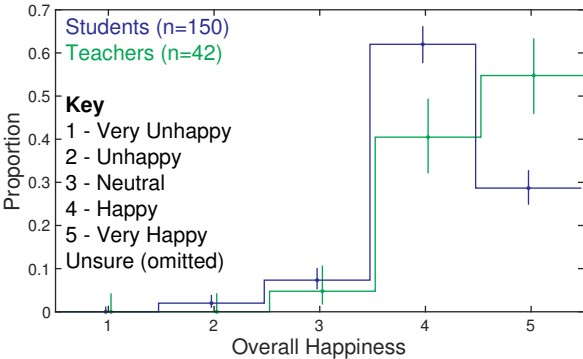

**Figure 3.** Distribution of students' (blue) and teachers' (green) overall happiness with their PRiSE projects. Error bars denote standard Clopper and Pearson (1934) intervals.

$p = 0.012$) and teachers ($3.80 \pm 0.20$, $p = 0.017$) than their respective means. No obvious trends were present by school metrics (explained in more detail in M.O. Archer, 2020).

While suggestive of extremely positive experiences with PRiSE, one also needs to compare these distributions against the typical responses of students and teachers for schools STEM engagement programmes. We use the results of Vennix et al. (2017) as such a benchmark, which surveyed 729 high-school students and 35 teachers about 12 different STEM outreach activities in the USA and Netherlands. This comparison reveals that PRiSE seems to be perceived considerably more positively than usual by both students (benchmark average $3.66 \pm 0.01$, $p = 1 \times 10^{-15}$ in a one-sample Wilcoxon signed-rank test) and teachers (benchmark average $3.84 \pm 0.08$, $p = 1 \times 10^{-7}$).

Secondly, students ($n = 135$) were asked for adjectives describing their experience of the projects overall. They were free to use any words they wanted and were not given a pre-selected list. Teachers ($n = 38$) were similarly asked to indicate observations of their students' experience also. Since 2016 this has resulted in 88 unique adjectives, with both students and teachers typically writing 2–3 words each. We present the results as the word cloud in Figure 4, where students and teachers have been given equal prominence by normalising their counts by their respective totals. We have indicated by colour from which group(s) the words originated, generally showing a lot of agreement between students' thoughts and teachers' observations. The most cited adjectives were (in descending order) interesting, challenging, exciting, inspiring and fun, similar to those from the pilot (M.O. Archer, 2017), with the top two adjectives being significantly greater than the subsequent ones. While in the pilot stage only positive adjectives were expressed, since then a few negative experiences have been conveyed such as time consuming, frustrating and stressful. These constitute a small minority of experiences though ($6 \pm 1\%$) and in most cases the same students also listed positive adjectives apart from only four individuals.

The most common themes that emerged from students' ($n = 110$) responses to open questions about their experience were that they feel they learnt a lot (62 responses)

> *"We have learnt so many new things relating to the magnetosphere and waves and we have developed new skills."*

(Student 3, Xavier's Institute for Higher Learning, MUSICS 2016)





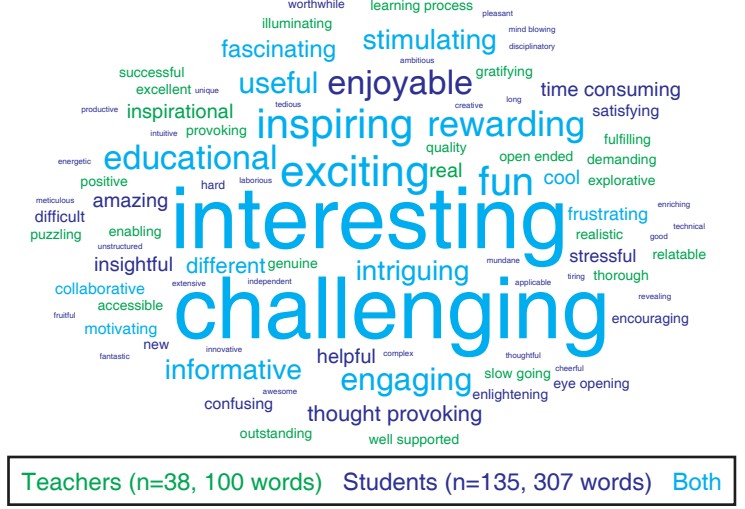

**Figure 4.** Word cloud of students' experiences. Colours indicate words identified by students (blue), teachers (green), or both (cyan). Students and teachers have been given equal total weight.

"*I learned so much! I would recommend it to all the younger kids at my school!*" (Student 111, St Trinians, SCREAM 2019)

"*I definitely learnt many new and interesting things and it helped me to develop my understanding of particle physics while aiding my A-Level knowledge.*" (Student 140, Jedi Academy, ATLAS 2020);

they found the projects' content and methods interesting (47 responses)

"*It brought my interests in programming, physics, maths and space together.*" (Student 100, Octavian Country Day School, PHwP 2019)

"*It has been very interesting to work with actual data and plan our own research project.*" (Student 116, St Trinians, SCREAM 2019)

"*It was very interesting to learn more in regards to astrophysics and the MUSICS project was a very safe space to do so. We got lots of support and it was fun.*" (Student 119, Pokémon Technical Institute, MUSICS 2020)

"*Coming into the Planet Hunting With Python project, my interests were mainly focused on the physics side of understanding brightness-curves and finding equations to solve for planetary parameters. However, in this project, my eyes were opened to the many uses of coding to analyse data, and it was a wonderful experience to learn about*

"*such an interesting area through a combination of theory and practical coding.*" (Student 120, Octavian Country Day School, PHwP 2020);

and they enjoyed the style of working in and with research which differs from their regular school experience (33 responses)





*"It was quite interesting to broaden our views and experience high level education."* (Student 90, Hill Valley High School, ATLAS 2019)

*"I enjoyed the opportunity to do science instead of just learning it."* (Student 101, Octavian Country Day School, PHwP 2019)

*"It was very nice to work with friends and work together to produce something."* (Student 106, Boston Bay College, PHwP 2019)

*"It was very fun to do our own research and I appreciated that help was always available even thought it is very*
*independent. It also shows how challenging research can actually be but also how rewarding it is once you start making progress."* (Student 143, Sunnydale High School, MUSICS 2020)

Neutral or negative experiences tended to be due to students finding the projects' content or open-ended way of working difficult or confusing (7 responses)

*"I did not understand most of the project or what I was supposed to do."* (Student 147, Jedi Academy, ATLAS
625 2020)

The vast majority of students seemed to ultimately enjoy this challenge though

*"The project gave us a lot of freedom and challenged us to think in different ways."* (Student 122, Jedi Academy, ATLAS 2020)

*"[It] made me more willing to take a go at challenges and what I deem hard."* (Student 130, Bending State College,
PHwP 2020)

and teachers agreed that the learning curve involved with the projects was advantageous for students

*"When the students got the hang of it they really enjoyed it."* (Teacher 16, Tree Hill High School, MUSICS 2018)
*"The students found it hard to identify what to do a project on and would have liked guidance on that, but I felt this was a good experience. They have developed grappling with open ended and difficult material."* (Teacher 23,
Smeltings, ATLAS 2019)

Teachers' feedback on their experience ($n = 34$) tended to praise how the projects allow their students to access and explore beyond the curriculum (29 responses)

*"A great framework to explore physics beyond the syllabus but still accessible."* (Teacher 26, Octavian Country Day School, PHwP 2019)

*"Excellent project that is open-ended allowing students to take it where they want."* (Teacher 27, Hogwarts, SCREAM 2019)

*"It linked nicely to some A-level topics but also felt like real science at university."* (Teacher 36, Starfleet Academy, MUSICS 2020)

*"This year I had an extremely motivated, enthusiastic and well-organised group of 7 students who fully immersed*



*themselves into the project and quickly took it in a direction outside my own understanding of this area of science.*
*This is exactly the experience I wanted them to have, and they were able to discover some genuinely novel pro-*
*cesses that had not been observed before - the hallmark of great scientific research!*" (Teacher 44, Sunnydale High
School, MUSICS 2020)

Therefore, both quantitative and qualitative data suggest students and teachers had highly positive and rewarding experiences
participating in PRiSE projects.

## 4.2 Support and resources

We originally asked students whether they felt they had received adequate support, finding overall positive results on a 5-point
Likert scale (M.O. Archer, 2017). However, students' qualitative responses explaining their answers often revealed a conflation
of the support provided by Queen Mary with that offered by their teacher. Therefore, from 2019 onwards we explicitly separated
these two aspects. Students ($n = 68$) were asked "Do you feel that support from your teacher was provided/available during the
project?" which yielded the following results: Strongly Agree (30), Agree (34), Neither Agree or Disagree (3), and Disagree
(1). The average response is $4.37 \pm 0.08$, which is considerably greater than the benchmark on teacher support reported by
Vennix et al. (2017) of $3.60 \pm 0.03$ ($p = 4 \times 10^{-10}$). Students' comments explaining their ratings ($n = 56$) revealed that teachers
provided them with advice, encouragement, and enthusiam (49 responses)

"*He always seemed fascinated by particle physics, nothing better than having a teacher as interested in a subject*
*as you are.*" (Student 128, Martha Graham Academy, ATLAS 2020)
"*My teacher has been very supportive and has helped us when we didn't understand something as well as encour-*
*aging us to taking a more innovative approach.*" (Student 124, Quirm College for Young Ladies, MUSICS 2020)
"*If we had a question, teachers were probably not useful. But if we did not know what to do or we were stuck, here*
*teachers were really useful and that was what we needed.*" (Student 145, Sunnydale High School, MUSICS 2020)

as well as arranging regular sessions for students to meet and visits or calls from the university when required (7 responses)

"*Our teacher arranged a Skype call with a professor from QMUL when we needed to ask questions about how*
*certain parts of the data were calculated.*" (Student 126, Harbor School, ATLAS 2020)
"[Our] *teacher would often ask us about it and hold meetings to catch up with us on our progress.*" (Student 143,
Sunnydale High School, MUSICS 2020)
"[Our] *teacher answered some of our questions and organised a weekly meet-up where students could ask each*
*other questions and work together.*" (Student 148, Jedi Academy, ATLAS 2020)

Neutral or negative responses tended to be explained by their teacher lacking specific knowledge about the research in response
to students' queries (2 responses)



> "*Although they were always ready to help, sometimes they didn't know the answers to our questions.*" (Student 122, Jedi Academy, ATLAS 2020)

> "[They] *didn't understand the content of the project.*" (Student 139, Jedi Academy, ATLAS 2020)

which is something we don't expect of teachers (cf. Shah and Martinez, 2016; Bennett et al., 2016, 2018), hence why support from the university is also offered.

Teachers' ($n = 18$) responses on a yes/no scale (chosen due to expected small number statistics) of whether they felt able to support their students were also highly positive with only 2 negative responses, a significant majority ($p = 0.001$ in a two-tailed binomial test). Bear in mind, however, that these responses were in light of the support provided from the university, something which a few teachers referenced in explaining their answers

> "*My own experience with research was handy but I felt that without this the students would still have been supported*" (Teacher 2, Hogwarts, SCREAM 2016)

> "*only through Martin's support*" (Teacher 4, Sweet Valley High School, MUSICS 2016)

> "*the teacher version of the handout was useful, but otherwise I could only give generic advice*" (Teacher 16, Tree Hill High School, MUSICS 2018)

Several teachers raised pressure on their time, with one teacher using this to justify their negative response

> "*sheer time pressure - big limiting factor*" (Teacher 13, St Trinians, SCREAM 2017)

another quoting this somewhat limited their ability to support students

> "*not as much as I would have liked (lack of available time)*" (Teacher 3, Xavier's Institute for Higher Learning , MUSICS 2016)

but most saying it was manageable

> "*The autonomous group work, with very little input from me, was great to see*" (Teacher 1, Hogwarts, SCREAM 2015)

> "*Students were quite self-sufficient so if I made suggestions they were able to do the leg work*" (Teacher 15, Spence Academy for Young Ladies, MUSICS 2018)

> "*The students were able to use the resources to self-manage their project*" (Teacher 19, Boston Bay College, PHwP 2019)

Teachers' ability and confidence in supporting the projects was another theme that emerged. Even with the teacher-specific resources provided, some felt they did not have the specific knowledge or skills to support the projects

> "[Unable to support due to a] *lack of knowledge of Python*" (Teacher 19, Boston Bay College, PHwP 2018)

Other teachers reflected that, similarly to their students, they too had experienced a learning curve through their involvement





"[I found it] *difficult at first*" (Teacher 8, Coal Hill School, MUSICS 2017)

ultimately becoming more determined and confident with time and in subsequent years

"*First time we've done this — I will do better next time*" (Teacher 17, Sunnydale High School, MUSICS 2018)

"*Second year that I ran it I feel more confident*" (Teacher 21, Hogwarts, SCREAM 2018)

which is further backed up by teachers' reported impacts and schools' significant repeated buy-in to the programme (M.O. Archer,
2020; M.O. Archer and DeWitt, 2020). The final theme raised was that for successful participation teachers believed the students needed the external motivation coming from the university rather than having project delivery being solely teacher-driven

"*Dr Archer was a great external lead to have. If I had been pushing them myself they would have taken it less seriously*" (Teacher 17, Sunnydale High School, MUSICS 2018)

Therefore, the comments from both students and teachers indicate that teachers alone would likely not have been able to
715 successfully support these research projects in their schools without both the resources and external motivation/mentoring provided as part of the PRiSE framework.

We now consider the specific elements of support. The evaluation of PRiSE's pilot recommended that numerous modes be provided as good practice for 'research in schools' schemes (M.O. Archer, 2017). From 2019 onwards we investigated participants' thoughts on each of the various aspects offered. Students ($n = 68$) and teachers ($n = 23$) were asked to rate the
720 usefulness of these as either 'unimportant', 'helpful', 'essential', or 'unsure'. This was chosen over a 5-point Likert scale due to an expected low number of responses, particularly from teachers. Any unsure or blank responses are neglected yielding 326 (out of a potential 408) student and 156 (out of 161) teacher responses. We divide these responses into negatives ('unimportant') and positives ('helpful' or 'essential'), though we acknowledge some may consider the 'helpful' response as neutral and thus our analysis takes both interpretations into account. The results are displayed in Figure 5 for the individual elements as well
as overall results obtained from totalling all responses. Both students and teachers overall rated the elements positively — coding the responses to values of 1 (negative) to 3 (essential) the overall means were $2.62 \pm 0.04$ for teachers and $2.23 \pm 0.03$ for students. The majority of teachers tended to give 'essential' ratings to most aspects and while these majorities are not statistically significant in a two-tailed binomial test, the average value for each element was greater than 2 to high confidence ($p < 0.002$ in one-sample Wilcoxon signed-rank tests). Students, on the other hand, mostly rated each element as 'helpful' as
well as stating slightly more negative responses than teachers, though again all elements' mean scores (apart from the kick-off workshop at $2.15 \pm 0.08$) were significantly greater than 2 ($p < 0.023$). While there are some variations in scoring amongst the different support elements, such as students and teachers respectively rating researcher visits and communications as the most essential, these differences to each group's overall results are slight and not statistically significant. One interpretation of this might be that most respondents answered unreflectively, ticking the same boxes for each item. However, no students and only
3 teachers gave the same answer in every category. This therefore suggests that all of the elements of support provided as part of PRiSE are almost equally important and necessary. This has been further elaborated on in teacher feedback:





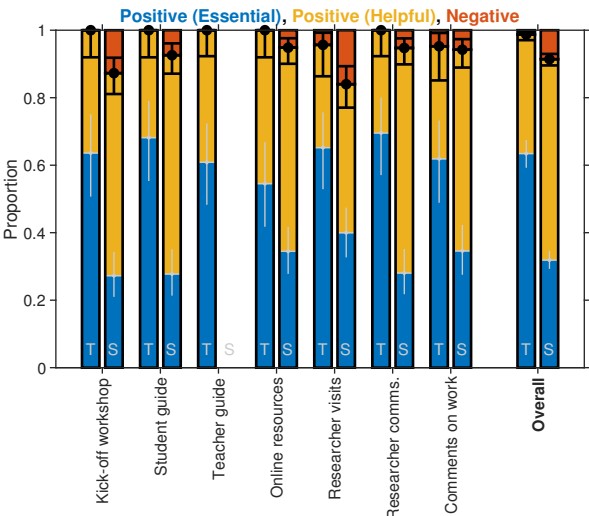

**Figure 5.** Usefulness of support provided to teachers (T, $n = 23$) and students (S, $n = 68$). Results are divided (black lines and associated error bars) into negative (red) and positive responses, with the latter subdivided (grey lines and error bars) into 'essential' (blue) and 'helpful' (yellow) elements. Error bars denote standard Clopper and Pearson (1934) intervals.

"*It is very well set up and open ended and the support received is magnificent.*" (Teacher 33, Boston Bay College, PHwP 2020)

"*Martin's guide to help the students was a very good balance of useful guidance and allowing them to find their own way through.*" (Teacher 38, Royal Dominion College, MUSICS 2020)

"*Truly excellent support from the Queen Mary team! They have visited us multiple times and have been so generous with their time. Students have learnt a great deal from them!*" (Teacher 44, Smallville High School, PHwP 2020)

## 5  Feedback from the university sector

Because we think there is potential for the PRiSE framework to spread beyond QMUL and be applied to other institutions' own areas of physics (and perhaps even STEM more generally) research, we wanted to assess how it is perceived by those from the university sector with interests and/or expertise in schools engagement. Feedback from our partner organisations seemed promising, for example with the South East Physics Network including PRiSE in their public engagement strategy (SEPnet, 2017), several of their member institutions expressing interest in adopting it, and the University of Surrey having already developed a pilot project. However, for a slightly more balanced perspective we collected evaluative data via an anonymous interactive online survey (see Appendix C) from researchers and public engagement professionals during a session at Interact 2019 (M.O. Archer, 2019), a day-long UK symposium concerning public engagement in the physical sciences.

The respondents, while heavily bought into schools engagement, tended to only undertake one-off activities (as detailed in Appendix C). After presenting the PRiSE framework to them, when asked on a 5-point Likert scale whether they ($n = 19$)





would now considering deeper approaches to outreach / engagement with schools, the results were: Strongly Agree (5), Agree
(9), Neither Agree or Disagree (5), and no responses in the two negative options. Coding these to a 1–5 scale yields a mean of
$4.0 \pm 0.2$, i.e. greater than neutral ($p = 1 \times 10^{-4}$ in a one-sample Wilcoxon signed-rank test).

In an open-ended question, participants were also asked to identify the main thing they had taken away from the session.
This yielded 17 responses. Through coding these answers three main themes emerged, with all responses covering only one
theme each. The first theme is about what types of schools engagement are possible (8 responses), with almost all thinking that
deeper programmes like PRiSE are achievable:

> "*Outreach does not have to just be workshop/talk based. It can be an interactive research based activity that
> supports research activities within HE*" (Participant 24)
> "*students are probably far more capable than schools and researchers might expect*" (Participant 16)
> "*maybe not as hard as I thought*" (Participant 18)

and only one person claiming that such approaches are not practical

> "*Would need a huge amount of time to set up something good - even with input from other people!*" (Participant 3)

The second theme (5 responses) concerns practical aspects towards delivering deeper programmes:

> "*lots of practical and multifaceted suggestions people in a variety of contexts can take and adapt for themselves*"
> (Participant 13)
> "*try and use existing resources available rather than reinventing the wheel*" (Participant 15)

The final theme (5 responses) is about evaluation methods and considerations of potential impacts, further explored for PRiSE
in M.O. Archer and DeWitt (2020), from engagement with schools:

> "*a relationship with professional researchers is of paramount importance in demonstrating the impact of an out-
> reach intervention*" (Participant 5)
> "*measuring impact by the teacher return rate*" (Participant 2)

These results suggest researchers and engagement professionals may be receptive to adopting the PRiSE framework, though
evidence of action following these immediate attitudes is really needed. We also acknowledge that this was a rather small
survey from a group already highly bought-in to schools engagement, thus results would likely be less positive from a wider
and more representative sample of all researchers. These are avenues which could be explored further in the future to gain a
better perspective on whether the PRiSE framework could realistically be rolled out further.

## 6 Conclusions

We have introduced a scalable framework for 'research in schools', open-ended independent research projects based in current
STEM research, called 'Physics Research in School Environments' (PRiSE) which aims to contribute towards increasing and





widening the uptake of physics (and more broadly STEM) in higher education. The theory of change behind the programme
has presented how, based on recent educational research and recommendations, PRiSE might be able to support participating
14–18 year-old students' existing science identities and enhance, or at least maintain, physics aspirations to help transform
these into degree subject destinations — a key issue for students at this stage of their educational journey (cf. Davenport et al.,
2020; L. Archer et al., 2020a). It has also detailed how, through working with teachers and schools, the programme could help
in ensuring school environments are able to support and nurture the science capital of all their students, thereby potentially
benefitting wider cohorts of school students (IOP, 2014).

The ethos behind PRiSE is to transform current scientific research methods, making them accessible and pertinent to a
diverse range of school students so that they can experience, explore, and undertake open-ended scientific research themselves.
Provision within the various emerging models of 'research in schools' projects has been little explored to date and in the wider
area of independent research projects it varies considerably in the level of support provided to schools (Bennett et al., 2016,
2018). We have, therefore, described in detail the considerations made in developing and evolving the PRiSE framework for
'research in schools' projects. These include a suite of interventions and resources to provide expert support and mentorship
from active researchers, which aim to enable a wide range of students, teachers, and schools to be able to participate. Our
approach attempts to find a balance (given necessarily limited time and resources) between reaching a large number of schools
and ensuring those schools are supported. It has enabled 1,326 students and 88 teachers across 67 schools to be involved since
2014, which is considerable compared to other similar programmes.

Feedback from participants upon completion has been very positive over the last 6 years, even compared to benchmark
results on schools engagement programmes with STEM in general. Students and teachers have found the projects of great
interest and have relished the challenge of working differently to in their regular school experience. They find the numerous
elements of support and interventions provided, uncommon in general with other schemes, as equally valued and necessary for
their participation. However, there is some attrition within the programme, which is to be expected and has been explored in
M.O. Archer (2020) showing that drop-off does not appear to be patterned by typical societal biases. Currently we have little
data on the experience of students and teachers that have dropped out of the programme, which is something that could be
explored in the future.

Researchers and public engagement professionals seem receptive to the PRiSE framework and it is slowly beginning to
spread to other institutions. This potential expansion might allow an assessment of how generally applicable the framework
is outside of its current London location and what other affordances might be required in these contexts. While PRiSE has so
far only concerned areas of physics research, 'research in schools' in general already span all the sciences (e.g. Bennett et al.,
2016, 2018; IRIS, 2020). We therefore see no reason why PRiSE's approach could not also be broadened to other STEM areas,
particularly areas of research based in data and/or analysis. We encourage researchers, and the public engagement professionals
who facilitate their activity, to consider adopting this way of working and hope this paper can inform this practice. In such cases,
it is recommended that PRiSE projects be embedded as core schools engagement activity within research groups. We would
be happy to support groups in developing, delivering, and evaluating pilot PRiSE projects around their own research, thereby
making use of the learning that has arisen from the programe over the last 6 years.





| Key Stage | Year | Final Exam | Age | Policy | PRiSE involvement |
|---|---|---|---|---|---|
| KS3 | 7 | None | 11–12 | Compulsory | Not recommended |
| KS3 | 8 | None | 12–13 | Compulsory | Not recommended |
| KS3 | 9 | None | 13–14 | Compulsory | Not recommended |
| KS4 | 10 | None | 14–15 | Compulsory | Select recommended |
| KS4 | 11 | GCSE | 15–16 | Compulsory | Select recommended |
| KS5 | 12 | AS-Level (optional) | 16–17 | Optional | All recommended |
| KS5 | 13 | A-Level | 17–18 | Optional | All recommended |

**Table A1.** Summary of the stages of secondary education in the English system.

## Appendix A: Information about UK/English schools

To those unfamiliar with the UK/English education system system, we provide some further notes here. The curriculum is broken down into Key Stages of duration 2–4 years, with those for secondary schools displayed in Table A1. The final two Key Stages culminate in GCSE and A-Level examinations respectively, with the latter being optional as education post-16 is not compulsory. Our recommendations to teachers about which year groups we recommend be involved with PRiSE are also highlighted in the table.

## 825 Appendix B: Participant evaluation questions

Here we list the questions posed in questionnaires that are considered within this paper, giving details of what phrasing was used, how participants could respond, and which years the question was posed. Follow-on questions are indicated by indentation and a down-right arrow (↳). Students' responded to the following:

| Question | Response type | Year(s) |
|---|---|---|
| Have you been happy with the research project overall? | 5-point Likert | 2016–2020 |
| ↳Please tell us why / why not | Open Text | 2016–2020 |
| What adjectives would you use to describe your experience of the project overall? | Open Keywords | 2016–2020 |
| Did you feel that support from your teacher was provided/available during the project? | 5-point Likert | 2019–2020 |
| ↳Please tell us why / why not | Open Text | 2019–2020 |
| Did you find the following elements of support from Queen Mary useful in supporting you? | Closed Options | 2019–2020 |
| How could we improve future projects? | Open Text | 2016–2020 |
| Any other comments? | Open Text | 2016–2020 |





The questions asked of teachers were:

| Question | Response type | Year(s) |
|---|---|---|
| Have you been happy with the research project overall? | 5-point Likert | 2016–2020 |
| ↳Please tell us why / why not | Open Text | 2016–2020 |
| What adjectives would you use to describe your students' experience of the project overall? | Open Keywords | 2016–2020 |
| Did you feel able to provide support to your students during the project? | Yes/No | 2016–2018 |
| ↳Please tell us why / why not | Open Text | 2016–2018 |
| Did you find the following elements of support from Queen Mary useful in supporting you and your students? | Closed Options | 2019–2020 |
| ↳Are there any ways we could further help you support your students with project work? | Open Text | 2019–2020 |
| Would you be interested in running this project or a similar one with us again? | Yes/No | 2016–2018 |
| ↳Please tell us why / why not | Open Text | 2016–2018 |
| How could we improve future projects? | Open Text | 2016–2020 |
| Any other comments? | Open Text | 2016–2020 |

## Appendix C: University sector questionnaire

The following questions were posed to university researchers and engagement professionals via on online interactive survey during a session at the 2019 Interact symposium (M.O. Archer, 2019).

| Question | Response type |
|---|---|
| What do you want your outreach / engagement with schools to achieve (i.e. what impact)? | Open Text |
| What sorts of outreach activities with schools do you do? (Select all that apply) | Closed Options |
| I am now considering deeper approaches to outreach / engagement with schools (Select only one) | 5-point Likert |
| What is the main thing you have taken away from this session? | Open Text |

For context on these participants, Figure C1 shows the distribution of the types of activity they undertake where they could select from:

A. Stall/stand: drop-in activities for schools at STEM or careers fairs

B. Talk: a typically one/two lesson slot featuring a predominantly one-way interaction

C. Workshop: a typically one/two lesson slot with mostly two-way interaction and often hands-on activities for students



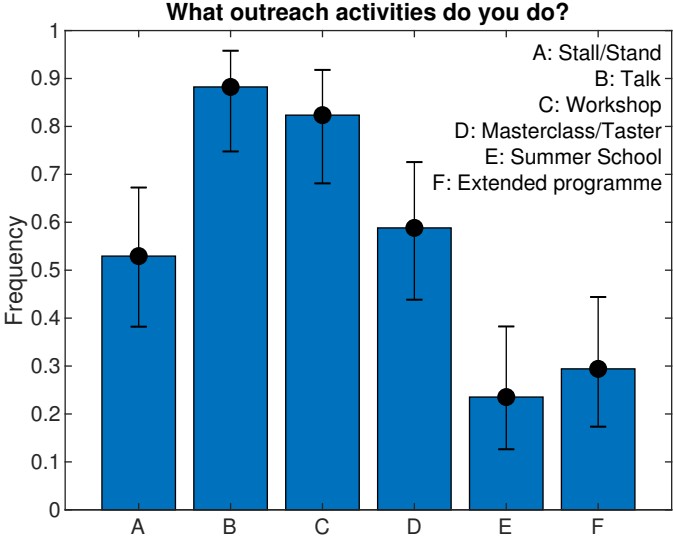

**Figure C1.** Bar chart of types of schools outreach activities undertaken by university researchers and engagement professionals ($n = 17$). Error bars denote the standard Clopper and Pearson (1934) interval.

D.  Masterclass/taster: half-day or day-long activities which may be comprised of talks and/or workshops

      E.  Summer school: several-day to week-long activities often involving some project work as well as talks and/or workshops

      F.  Extended programme: multiple interventions with the same group of students over a protracted period of time

Unsurprisingly, one-off activities such as talks and workshops scored the highest whereas more the protracted engagements, summer schools and extended programmes, were significantly ($p < 0.0019$) less common. The attendees were also asked what
they hoped to achieve (i.e. the aims or intended impacts) through their school engagements via an open question. Performing a thematic analysis of the qualitative results, it was possible to categorise the majority of answers into the following:

     – Change school students' aspirations (9 people), with the word "inspire" often used

     – Enhance students' awareness or understanding of STEM (6 people), often in the context of primary research

     – Tackle societal biases in STEM (4 people), most often gender

Note that some responses covered more than one of these aims. Other stated motivations outside of these themes included "access to a student population for [research] studies", to "build relationships", and to deliver "meaningful content". The three themes are in general agreement with those determined by Thorley (2016) in a larger survey of UK physicists.



*Data availability.* Data supporting the findings of this study that is not already contained within the article or derived from listed public domain resources are available on request from the corresponding author. This data is not publicly available due to ethical restrictions based on the nature of this work.

*Author contributions.* MOA conceived the programme and its evaluation, performed the analysis, and wrote the paper. JDW and CT contributed towards the analysis, validation, and writing.

*Competing interests.* The authors declare that they have no conflict of interest.

*Acknowledgements.* We thank Dominic Galliano and Olivia Keenan for helpful discussions. MOA and CT are grateful for funding from the Ogden Trust. This programme has been supported by a QMUL Centre for Public Engagement Large Award, and STFC Public Engagement Small Award ST/N005457/1.



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
