# Peer review of "Evaluating participants' experience of extended interaction with cutting-edge physics research through the PRiSE 'research in schools' programme"

_Geoscience Communication, 2020_

## Referee Comment (RC1) · Anonymous Referee #1 · 25 Aug 2020

The article submitted about the PRiSE project is very interesting. The quality of the presentation is excellent ... the implementation and results of the project are clearly presented. Nevertheless, I would have liked to have seen more details for the teachers involved in the project. What strategy is actually implemented for teacher training during the PRiSE project?

line 172 > 'Therefore, opportunities for teachers' development are integrated within the programme rather than being a separate offering to schools'

---

## Author Comment (AC1) · 7 Sep 2020

**The article submitted about the PRiSE project is very interesting. The quality of the presentation is excellent ... the implementation and results of the project are clearly presented. Nevertheless, I would have liked to have seen more details for the teachers involved in the project. What strategy is actually implemented for teacher training during the PRiSE project?**

**line 172 'Therefore, opportunities for teachers' development are integrated within**

[Figure]

**the programme rather than being a separate offering to schools'**

We thank the reviewer for taking the time to assess the manuscript and for their report. We agree that expanding discussion of our strategy and implementation for teacher development within PRiSE would improve the manuscript.

We will add an overview of our integrated approach after line 273 as follows:

> Teachers' involvement at all stages also presents opportunities to them for continuing professional development. This is implemented informally and integrated within the programme in the form of both bespoke resources and ongoing dialogues between teachers and researchers. These are aimed to enhance teachers' knowledge about the projects' underlying science and how they link to curriculum-based topics where appropriate, their skills and confidence surrounding current research topics and methods, and their pedagogy in mentoring independent project work.

We will expand in section 2.4.1 for each intervention stage specific considerations for teachers:

- Kick-off: The outreach officer will also have an informal chat with (particularly new) teachers concerning how to go about undertaking and supervising the project, answering any questions or concerns they may have with either the science, activities, or project management.

- Prescribed work: Students are still required to problem solve throughout these stages and we purposely do not provide them with all the answers to prompt this, though teachers are given guidance in their resources to support student efforts.

- Visit: Teachers are encouraged to participate in these meetings and additionally further informal chats (similar to those taking place during the kick-off) between

the researcher and teacher occur to help with their project supervision and continuing professional development.

- Independent project: Potential research questions are suggested in the guides provided, with further advice for teachers on these being given in their versions, and students' ideas are discussed during visits and/or webinars.

On line 378 we will add that by having teachers act as intermediaries for students' ad hoc support this also forms part of teachers' development:

This is done not only for logistical and safeguarding reasons, but also provides further opportunities for university–teacher dialogues that can contribute to their professional development.

These additions along with the discussion of the contents of teacher guides already present in the manuscript (lines 404-407) should give readers a better idea of our approach to teacher development through the programme.

---

## Referee Comment (RC2) · Anonymous Referee #1 · 8 Sep 2020

I would like to thank you for taking my remarks into account. I find that, in your paper now, the teachers' training in the framework of PRiSE is more explicit.We are well aware, as you say in your article, that the role of teachers is very important for the success of the projects ... and it is therefore important to ensure that teachers are well supported.

---

## Referee Comment (RC3) · Anonymous Referee #2 · 17 Nov 2020

Many thanks to authors for all of their efforts in putting together this interesting piece of research. I am sure that many of those working in Outreach and Public Engagement in HE could use some elements of this article for their own benefit. That said there are few elements I ask to authors to review. These comments should be seen as constructive and should really enhance the current structure of the article

1. The Theory of Change (ToC) presented reads well and it follows very closely the Theory of Change published by Davenport et al 2020., in terms of identified audiences, causal paths and terminology. I recommend a more explicit acknowledgement by the

authors to Davenport et al 2020. Where I think there is room for improvement is to explain or summarise and assumptions and barriers that often accompany a ToC, as throughout the paper some of these emerge (e.g - researchers or institutional buy-in a barrier to your ToC ). Finally the ToC description needs a bit more details. For instance what is the meaning of the different shades of the same colour

2. The framework as well described by the authors, feels disconnected from the ToC and more references throughout the text should be made to the ToC especially in section 2.3 and 2.4 For example line 230 to 245 , removal of barriers, involvement of teachers etc, really highlight that these are aspects of your ToC.

3. Even though the PRISE framework has been presented as scalable, what are the lessons learnt by the authors? what are the recommendations to other practitioners in the field? Adding a few paragraphs in the conclusions, or even some bullet points, would address not only the scalability of PRISE but the transferability of PRISE to another subject or institution (e.g. - produce a detailed guide for students and teachers, etc)

I believe that addressing these point would really enhance an already good pice of research.

---

## Referee Comment (RC4) · Anonymous Referee #3 · 18 Nov 2020

General comments

This paper makes a valuable contribution to the available literature on undertaking projects that involve school students in research. It is generally well-constructed and well written, leading the reader through the premise, structure and success of the programme. The detailed exposition of the workings of PRiSE is especially welcome as it facilitates the successful replication of such a programme without a duplication of the evidently extensive effort and multiple trials that have been required to bring the programme to its current level.

[Figure]

There are some points in the paper (as will be addressed in the specific comments below) that would benefit from further consideration; however, these comments are mostly fairly minor, and are noted in a spirit of bringing the level of every part of the paper to the high standard it exhibits overall.

The authors give a thorough depiction of the landscape in which this work sits, taking care to give details of other similar projects distinct from PRiSE. Nevertheless, as noted below, these other projects are not always considered in a positive light. It might be wise not to be over-critical at the risk of sounding petty rather than constructive. However, proper credit is given where appropriate, both to work outside this project and to the researchers and other staff involved in PRiSE, which was heartening to see.

The title seems fair, although there is an emphasis on 'destinations' that is less apparent within the body of the paper. Although this is mentioned within the section on the Theory of Change, there seems to be little further discussion or evidence of the destinations of students that take part in PRiSE. Nevertheless, the abstract provides a concise, complete and clear summary of the contents of the paper.

The language is largely fluent and precise. On occasion, some of the sentence structures are a little hard to follow on a first reading. In particular, there is substantial use of possessive apostrophes that on occasion impede initial comprehension. It may be worth reconsidering some of these to aid the flow of the text (as opposed to the text's flow).

The paper is well-referenced throughout, with many recent publications cited, demonstrating a laudable grasp of current best practice and educational research. This is to be highly commended.
* * *
Specific comments

The specific comments are given with line references relating to the pre-print (pdf) of

the paper.

- Line 9: 'with all elements appearing equally important.' – it would be useful (perhaps later in the article) to have a simple list of all the elements that are being encompassed by this phrase.

- Line 66 on: The discussions of other similar projects, while not obviously straying from factual, nevertheless read as ungenerous. E.g. line 73: 'other memberships are seemingly justified to ensure that schools make a commitment to working with the university'; line 88: 'While some researchers/academics have designed or consulted on some IRIS projects, they appear in general to have little involvement supporting students or teachers'. This could perhaps be construed as criticism of the other projects (with the aim of elevating PRiSE) which may or may not be considered constructive at this juncture.

- I understand that the Theory of Change as presented here is discussed in more detail in another publication. Nevertheless, I would query a couple of aspects that are here presented without substantial examination (though I recognise this review comment may not be the best home for this remark and the authors may feel that no response or alteration is merited.) ** Figure 1: The implication that 'Know other people interested in physics' leads to 'See themselves as equals in physics to those from different backgrounds'. I don't know that this follows. I think you can quite easily know other people interested in physics and *not* see yourself as "equal in physics" to those other people. ** Line 163: 'By interacting first-hand with "real physics" through the projects and working with active researchers, students (especially those from under-represented groups) should feel included'. I think it is perfectly possible to do physics research and yet feel excluded. I am not convinced that under-represented students will automatically feel included, simply by virtue of doing "real physics", especially if they do not recognise themselves in the active researchers they are connected with or are a minority in the group taking on a project in their school.

- Line 201 -205. How do the other IRIS physics projects compare here? As it stands, it reads like a cherry-picked list of worst performers, highlighted to make PRiSE look good. If further data on the numbers of researchers / schools is not available for other IRIS projects, then this is worth noting here to avoid this impression.

- Line 196: It becomes apparent here that some schools have taken part but then dropped out. It might be worth pointing this out explicitly, and possibly signposting the later short discussion of this (e.g. around line 806)

- Line 240 – 242: teachers decide who to offer the PRiSE projects to. Do you have any thoughts on how successful teachers are at selecting students who excel on the projects?

- Line 248: 'We allow teachers to determine how best to integrate the projects within their school, though provide advice on this.' From the perspective of an outreach practitioner hoping to replicate the success of PRiSE, it would be interesting and useful to see this advice – perhaps included in an appendix?

- Line 286 on: How much drop-off do you typically see between teachers applying for projects and then not taking up an offer come the new academic year?

- Line 309: 'though this latter approach often proves unsuccessful' – thank you for including this kind of helpful detail

- Line 463: What is 'the UK coding agenda'? This phrase needs further explication and / or a reference

- Line 534: Feedback from the university sector. This is a bit confusing – it's a little unclear what the university sector is being asked or why, and how that connects with the previous discussion of participant feedback. Although there will be further detail given later, it might be worth clarifying some of it at this stage. Maybe it's simply the mention of "the workshop" (line 535) without context that is disconcerting.

- Line 589: I enjoyed the inclusion of the negative words in the word cloud, and appreciated that they were highlighted here.

- Line 791: 'The ethos behind PRiSE is to transform current scientific research methods' This could be read as though you are trying to alter the way the scientists undertake their research. Perhaps consider re-phrasing this, if that is not your intention.
* * *
Technical corrections

- Line 143: 'with standard one-off (or even short-series of) intervention(s)' – this doesn't quite read right to me

- Line 169: 'Experience from physics outreach officers . . . have shown' – grammar error. Should either be 'Experiences . . . have shown' or 'Experience . . . has shown'

- Line 174: 'the impacts of PRiSE can be felt much wider' – grammar. Suggest 'can be felt much more widely'.

- Line 183: 'another major influence on young people's aspirations are family' – grammar error. Should be 'another major influence . . . is family'.

- Line 213: 'One might think it is feasible that students' work on PRiSE projects contribute to novel research.' - Grammar: contributes - This sentence is generally hard to follow – consider revising

- Line 423: 'organic semiconducters' – typo: semiconductors

- Line 502: 'and responsibilities have remained largely been falling to only a few people per PRiSE project' – grammar. Remove 'remained'?

- Line 803: 'and have relished the challenge of working differently to in their regular school experience' – grammar. Remove 'in'?

---

## Referee Comment (RC5) · Anonymous Referee #4 · 19 Nov 2020

Thank you to the authors for working to summarize their program and research as part of the PRiSE program. It is clear that this group has taken the task of creating and evaluating their educational program seriously and I commend them on identifying many different facets of the program to document and share with the broader community. The paper is generally well presented but as a reader first learning about this program I have some major revisions to suggest.

Major Considerations for Revision: While I see the clear value and need to share this work with the broader community, especially given the authors' goal to "introduce a

scalable framework for protracted research-based engagement with schools", I have some questions about how this and the other papers submitted simultaneously in review in Geoscience Communication about the same program differ from one another and how they each meet the journal requirement of making "a substantial contribution to scientific progress within the scope of Geoscience Communication (substantial new concepts, ideas, methods, or data)".

Further, based on the references cited in this manuscript, the lead author also has another paper in review at another journal that seems to align with a similar premise being presented here. As a reader (who hasn't read all of these manuscripts to know exactly how they differ), I'm left wondering why someone would need to read four papers about a program to understand the structure and impact. While I fully appreciate the appropriateness of evaluating and interpreting results from a program like this in multiple ways, the current structure of the arguments suggests that they might be able to present their work in one well-structured and concise paper (or two) that really uses the data to substantiate the claims being made and demonstrates how they are meeting their Theory of Change which states: "The intended impact of PRiSE is to contribute towards the increased uptake and diversity of physics at higher education."

As a reader new to the program, and taking the abstract at face value, I found myself asking fundamental questions about the structure, resources, personnel and design of the program. This is touched on in brief in various parts of the paper but some challenges about the program structure are mentioned starting in line 500 that seem to warrant further comment, especially in the context of thinking about scalability of this program. Other details like the core resources or research that undergird the program are mentioned in the latter part of the paper and seem a bit out of place. Perhaps some of the text could be instead captured in visuals or a diagram? Overall, while the Theory of Change and surrounding literature review are helpful for framing the need and context of the PRiSE program, the body of the text and data presented don't seem to directly align with or support the premise of the paper as articulated in the abstract

and conclusions.

To concretely illustrate this, in the abstract the authors state "This illustrates that the model appears to provide highly positive experiences that are otherwise not accessible to schools and that the extraordinary level of support offered is deemed necessary with all elements appearing equally important. Researchers and public engagement professionals seem receptive to the PRiSE framework of schools engagement and it has started to spread to other institutions." While the authors present some data to demonstrate their programmatic success, for their most critical claims, they point the reader to a different paper (as above) and don't really touch on the focal point of their theory of change. Further, they mention on several occasions in the paper the "extraordinary level of support" needed and offered through this program by the researchers but do not elaborate on how this might be a barrier to the scalability of their program. It would be helpful to the readers if the authors were more explicit about how much time is required from researchers to support this type of programming, how researchers are recruited and rewarded/acknowledged for their participation and how the program itself is funded or supported, especially in light of the acknowledged barriers to sustaining engagement by researchers. These types of structural and programmatic details are key to seeing how the program supports their ToC and offers valuable insights for those seeking to recreate this type of 'research in schools' program.

Building on the other reviewer's comments, I think there is ample opportunity to streamline the text, and clarify the presentation to only those details most salient to communicating to the reader the design and implementation elements of the program, while being much more explicit about how the data they have collected demonstrate if/how (or not) the program 'meets' their Theory of Change. This is essential to demonstrate a) how the program is scalable and b) the documented value and impact and therefore, why it is a model that should be scaled to other schools/locations/programs. I would encourage the authors to significantly revise this manuscript and to think about how to present the details about how the program works and the data they have that indicates

that the program is successful (and why) together in one paper.

Based on the review criteria, this article falls short in demonstrating (in relation to the other papers submitted for review at the same time) how each makes a unique and substantial contribution that warrants publication, and as currently written, this paper does not really provide sufficient evidence to support the interpretations and conclusions. I have no doubt that through some more careful writing, streamlining of the text and analysis of the data alongside the programmatic structure, that readers would see the substantial contribution being made through this program and its structure and the value it offers as a model that could be replicated elsewhere.

While there is work to do, I really do commend the authors on their thoughtful approach, clear investment in data collection and analysis, and for developing and iterating on a program that seeks to make a novel contribution for bringing research to schools. They certainly have invested an enormous amount of time and dedication to the PRiSE program and I really hope to see this work shared with the science communication and education communities.

---

## Author Comment (AC2) · 2 Dec 2020

We thank the reviewer for their comments on the revisions and agree with their sentiment.

---

## Author Comment (AC3) · 2 Dec 2020

**Many thanks to authors for all of their efforts in putting together this interesting piece of research. I am sure that many of those working in Outreach and Public Engagement in HE could use some elements of this article for their own benefit. That said there are few elements I ask to authors to review. These comments should be seen as constructive and should really enhance the current structure of the article.**

[Figure]

We thank the reviewer for their time in reviewing the manuscript and have considered their comments carefully as follows.

**1. The Theory of Change (ToC) presented reads well and it follows very closely the Theory of Change published by Davenport et al 2020., in terms of identified audiences, causal paths and terminology. I recommend a more explicit acknowledgement by the authors to Davenport et al 2020. Where I think there is room for improvement is to explain or summarise and assumptions and barriers that often accompany a ToC, as throughout the paper some of these emerge (e.g - researchers or institutional buy-in a barrier to your ToC ). Finally the ToC description needs a bit more details. For instance what is the meaning of the different shades of the same colour.**

We will more explicitly highlight the Theory of Change of Davenport et al. (2020), which focuses more heavily on careers and interventions at primary and lower secondary school, and contrast it to that for PRiSE. We can add some short notes about basic assumptions made in the Theory of Change and then later in the paper, when describing the PRiSE framework, relate considerations and potential barriers back to the Theory of Change. We will also expand the caption of the Theory of Change to aid in its interpretation. The colours in the figure are for the different audiences considered and the shadings indicate the timeframe of theorised outcomes going from lighter (short- and medium-term) to darker (long-term and impact) with time running vertically from top to bottom.

**2. The framework as well described by the authors, feels disconnected from the ToC and more references throughout the text should be made to the ToC especially in section 2.3 and 2.4 For example line 230 to 245 , removal of barriers, involvement of teachers etc, really highlight that these are aspects of your ToC.**

We will make more reference to the Theory of Change when describing the considerations made in developing the PRiSE framework.

**3. Even though the PRISE framework has been presented as scalable, what are the lessons learnt by the authors? what are the recommendations to other practitioners in the field? Adding a few paragraphs in the conclusions, or even some bullet points, would address not only the scalability of PRISE but the transferability of PRISE to another subject or institution (e.g. - produce a detailed guide for students and teachers, etc)**

We will expand the discussion of PRiSE's scalability in section 2.3 with regards to balancing reach and impact. We will also add some more general recommendations to the conclusions for practitioners looking to establish 'research in schools' projects/programmes more generally, drawing from the results presented from the PRiSE approach.

**I believe that addressing these point would really enhance an already good piece of research.**

We agree with the reviewer that these points will enhance the manuscript.

---

## Author Comment (AC4) · 2 Dec 2020

**General comments**

This paper makes a valuable contribution to the available literature on undertaking projects that involve school students in research. It is generally well-constructed and well written, leading the reader through the premise, structure and success of the programme. The detailed exposition of the workings of PRiSE is especially welcome as it facilitates the successful replication of such a pro-

gramme without a duplication of the evidently extensive effort and multiple trials that have been required to bring the programme to its current level.

**There are some points in the paper (as will be addressed in the specific comments below) that would benefit from further consideration; however, these comments are mostly fairly minor, and are noted in a spirit of bringing the level of every part of the paper to the high standard it exhibits overall.**

We thank the reviewer for their time and comments on our manuscript. We have carefully considered all the reviewer's suggested improvements.

**The authors give a thorough depiction of the landscape in which this work sits, taking care to give details of other similar projects distinct from PRiSE. Nevertheless, as noted below, these other projects are not always considered in a positive light. It might be wise not to be over-critical at the risk of sounding petty rather than constructive. However, proper credit is given where appropriate, both to work outside this project and to the researchers and other staff involved in PRiSE, which was heartening to see.**

We will adjust the tone slightly in places when discussing other projects to mitigate any perceived negativity. The points raised summarise the information available about these projects, highlighting the need for more publications detailing the provision within this area, and also to be able to compare/contrast to the considerations made in developing PRiSE's approach.

**The title seems fair, although there is an emphasis on 'destinations' that is less apparent within the body of the paper. Although this is mentioned within the section on the Theory of Change, there seems to be little further discussion or evidence of the destinations of students that take part in PRiSE. Nevertheless, the abstract provides a concise, complete and clear summary of the contents of the paper.**

Upon reflection, we feel that the title perhaps does not best encapsulate the content of the manuscript. So we will change it to "Developing a framework to bolster school students' aspirations through extended interaction with cutting-edge research: 'Physics Research in School Environments'". Assessing whether PRiSE has actually affected the degree destinations of students is covered in the companion paper on impact (Archer and DeWitt, 2020). However, we will add more references back to this aim of the programme throughout the manuscript.

Archer, M. O. and DeWitt, J.: "Thanks for helping me find my enthusiasm for physics!" The lasting impacts research in schools projects can have on students, teachers, and schools, Geosci. Commun. Discuss., https://doi.org/10.5194/gc-2020-36, in review, 2020.

**The language is largely fluent and precise. On occasion, some of the sentence structures are a little hard to follow on a first reading. In particular, there is substantial use of possessive apostrophes that on occasion impede initial comprehension. It may be worth reconsidering some of these to aid the flow of the text (as opposed to the text's flow).**

Upon revision we will attempt to improve the language throughout.

**The paper is well-referenced throughout, with many recent publications cited, demonstrating a laudable grasp of current best practice and educational research. This is to be highly commended.**

We thank the reviewer for this comment.

**Specific comments**

**The specific comments are given with line references relating to the pre-print (pdf) of the paper.**

**- Line 9: 'with all elements appearing equally important.' – it would be useful (perhaps later in the article) to have a simple list of all the elements that are**

**being encompassed by this phrase.**

We will add a list to make these elements clearer.

**- Line 66 on: The discussions of other similar projects, while not obviously straying from factual, nevertheless read as ungenerous. E.g. line 73: 'other memberships are seemingly justified to ensure that schools make a commitment to working with the university'; line 88: 'While some researchers/academics have designed or consulted on some IRIS projects, they appear in general to have little involvement supporting students or teachers'. This could perhaps be construed as criticism of the other projects (with the aim of elevating PRiSE) which may or may not be considered constructive at this juncture.**

As mentioned earlier, we will adjust the tone of these statements. For example removing the phrase "seemingly justified" with regards to HiSPARC (line 73) and adding that IRIS itself acts as the main point of contact for schools rather than researchers (line 88). These points are important to include in order to contrast the different approaches currently used and in explaining the considerations made in developing PRiSE's approach later in the manuscript.

**- I understand that the Theory of Change as presented here is discussed in more detail in another publication. Nevertheless, I would query a couple of aspects that are here presented without substantial examination (though I recognise this review comment may not be the best home for this remark and the authors may feel that no response or alteration is merited.) ** Figure 1: The implication that 'Know other people interested in physics' leads to 'See themselves as equals in physics to those from different backgrounds'. I don't know that this follows. I think you can quite easily know other people interested in physics and *not* see yourself as "equal in physics" to those other people. ** Line 163: 'By interacting first-hand with "real physics" through the projects and working with active researchers, students (especially those from underrepresented groups) should**

**feel included'. I think it is perfectly possible to do physics research and yet feel excluded. I am not convinced that under-represented students will automatically feel included, simply by virtue of doing "real physics", especially if they do not recognise themselves in the active researchers they are connected with or are a minority in the group taking on a project in their school.**

We appreciate the reviewer's point that by no means are all the causal links presented in the Theory of Change guaranteed to occur. Indeed only through identifying assumptions and potential barriers to these links can a successful programme be formulated which might go on to enact the theorised outcomes and impacts. We will add this point to the manuscript to make this clearer. Furthermore, we will add more references back to the Theory of Change when discussing the considerations made in developing PRiSE. As the reviewer mentions, whether PRiSE has been successful in these aspects is assessed in another publication as is stated at the end of this section (lines 188-189).

**- Line 201 -205. How do the other IRIS physics projects compare here? As it stands, it reads like a cherry-picked list of worst performers, highlighted to make PRiSE look good. If further data on the numbers of researchers / schools is not available for other IRIS projects, then this is worth noting here to avoid this impression.**

The reviewer is correct in that we simply do not have any concrete information on the other IRIS projects and so will make a note of this here as suggested.

**- Line 196: It becomes apparent here that some schools have taken part but then dropped out. It might be worth pointing this out explicitly, and possibly signposting the later short discussion of this (e.g. around line 806)**

We thank the reviewer for this suggestion which we will explicitly highlight here.

**- Line 240 – 242: teachers decide who to offer the PRiSE projects to. Do you have**

**any thoughts on how successful teachers are at selecting students who excel on the projects?**

Unfortunately, we don't have any specific information on how teachers go about selecting students, which we will raise in the manuscript. However, we can also note that the average number of students per school each year is around 12, which compared to the national average class size in A-Level physics of 16 (RAE et al., 2015) indicates teachers involve a significant majority (or in many cases the entirety) of their cohorts in PRiSE.

RAE, IOP, and Gatsby Foundation: School sixth forms with no entries for A-level physics, Tech. rep., Institute of Physics, https://www.iop.org/about/publications/school-sixth-forms-no-entries-level-physics, accessed: Aug 2020, 2015.

**- Line 248: 'We allow teachers to determine how best to integrate the projects within their school, though provide advice on this.' From the perspective of an outreach practitioner hoping to replicate the success of PRiSE, it would be interesting and useful to see this advice – perhaps included in an appendix?**

So far this has been done through informal discussions between the outreach officer and teachers during the kick-off meeting, which we will note as:

> The outreach officer will also have an informal chat with (particularly new) teachers concerning how to go about undertaking and supervising the project, answering any questions or concerns they may have with either the science, activities, or project management.

We had hoped to additionally formalise this aspect into the planned 'how to' guides for teachers, but as noted on lines 416-419 we have not had time to co-create these yet so they cannot be included as an appendix.

**- Line 286 on: How much drop-off do you typically see between teachers applying**

**for projects and then not taking up an offer come the new academic year?**

There is a 33±5% drop out between application/assignment and the initial kick-off meeting in the new academic year. Retention within the programme is explored fully in one of the companion papers submitted (M.O. Archer, 2020) which we will highlight.

Archer, M. O.: School students from all backgrounds can do physics research: On the accessibility and equity of the PRiSE approach to independent research projects, Geosci. Commun. Discuss., https://doi.org/10.5194/gc-2020-37, in review, 2020.

**- Line 309: 'though this latter approach often proves unsuccessful' – thank you for including this kind of helpful detail**

We thank the reviewer for this comment.

**- Line 463: What is 'the UK coding agenda'? This phrase needs further explication and / or a reference**

We will add a short note and reference (e.g. https://www.gov.uk/government/news/schools-minister-announces-boost-to-computer-science-teaching) to highlight the UK government's desire for more young people to develop computer programming skills.

**- Line 534: Feedback from the university sector. This is a bit confusing – it's a little unclear what the university sector is being asked or why, and how that connects with the previous discussion of participant feedback. Although there will be further detail given later, it might be worth clarifying some of it at this stage. Maybe it's simply the mention of "the workshop" (line 535) without context that is disconcerting.**

We thank the reviewer for highlighting the need for a brief discussion of the content of the workshop, which we will add.

**- Line 589: I enjoyed the inclusion of the negative words in the word cloud, and**

**appreciated that they were highlighted here.**

It is important that the entirety of the collected data are presented and discussed in a balanced and appropriate way, which we have aimed to do throughout.

**- Line 791: 'The ethos behind PRiSE is to transform current scientific research methods' This could be read as though you are trying to alter the way the scientists undertake their research. Perhaps consider re-phrasing this, if that is not your intention.**

We will rephrase this to "The ethos behind developing PRiSE projects is to transform current scientific research methods, finding ways to make these methods accessible and pertinent to a diverse range of school students so that students can experience, explore, and undertake open-ended scientific research themselves."

**Technical corrections**

**- Line 143: 'with standard one-off (or even short-series of) intervention(s)' – this doesn't quite read right to me**

We will alter this to "with standard one-off interventions, or even short-series, showing no real changes"

**- Line 169: 'Experience from physics outreach officers . . . have shown' – grammar error. Should either be 'Experiences . . . have shown' or 'Experience . . . has shown'**

We will make this correction.

**- Line 174: 'the impacts of PRiSE can be felt much wider' – grammar. Suggest 'can be felt much more widely'.**

We will make this correction.

**- Line 183: 'another major influence on young people's aspirations are family' –**

**grammar error. Should be 'another major influence . . . is family'.**

We will make this correction.

**- Line 213: 'One might think it is feasible that students' work on PRiSE projects contribute to novel research.' - Grammar: contributes - This sentence is generally hard to follow – consider revising**

We will change this to "One might think that students' work on PRiSE projects can contribute to novel research".

**- Line 423: 'organic semiconducters' – typo: semiconductors**

We will make this correction.

**- Line 502: 'and responsibilities have remained largely been falling to only a few people per PRiSE project' – grammar. Remove 'remained'?**

We will make this correction.

**- Line 803: 'and have relished the challenge of working differently to in their regular school experience' – grammar. Remove 'in'?**

We will make this correction.

---

## Author Comment (AC5) · 15 Dec 2020

**Thank you to the authors for working to summarize their program and research as part of the PRiSE program. It is clear that this group has taken the task of creating and evaluating their educational program seriously and I commend them on identifying many different facets of the program to document and share with the broader community. The paper is generally well presented but as a reader first learning about this program I have some major revisions to suggest.**

[Figure]

We appreciate the reviewer's attention to our manuscript and are particularly grateful for raising our awareness of areas that are not as clear as we intended them to be. We have carefully considered the reviewer's points and respond to them below.

**Major Considerations for Revision: While I see the clear value and need to share this work with the broader community, especially given the authors' goal to "introduce a scalable framework for protracted research-based engagement with schools", I have some questions about how this and the other papers submitted simultaneously in review in Geoscience Communication about the same program differ from one another and how they each meet the journal requirement of making "a substantial contribution to scientific progress within the scope of Geoscience Communication (substantial new concepts, ideas, methods, or data)"**

The demarcation of the different papers submitted to Geoscience Communication are outlined on lines 115-117, and concern three separate areas:

- This paper: An introduction to the framework of this programme, the considerations made in its provision and support, and how they have been perceived by stakeholders (so-called process evaluation).

- Impact evaluation exploring the actual benefits for students and teachers that might have resulted from the programme.

- Audience evaluation assessing the diversity, accessibility and equity within the programme for the schools we work with.

Each paper contains clear and separate conclusions. Other published papers in this field by other authors, for example the various papers cited on IRIS's programme, also split the content and focus up in similar ways thus we feel this is appropriate.

**Further, based on the references cited in this manuscript, the lead author also has another paper in review at another journal that seems to align with a similar premise being presented here.**

The other paper that the reviewer refers to is an in-press landscape review of various engagement programmes that use repeated interventions rather than one-offs and thus features a wide range of other programmes. We will highlight this in the manuscript for clarity.

**As a reader (who hasn't read all of these manuscripts to know exactly how they differ), I'm left wondering why someone would need to read four papers about a program to understand the structure and impact.**

How the papers differ in their focuses is outlined on lines 107-117 but we will try to clarify the differences between the three papers. The programme is significantly more extended, consisting of several different parts, than typical one-off engagement approaches published in this journal and others, thus it is not surprising that a full exploration of its structure and impact is lengthy. (As noted, the fourth is a review of a range of programmes and reading it is not necessary to understand the structure and impact of this particular programme.)

**While I fully appreciate the appropriateness of evaluating and interpreting results from a program like this in multiple ways, the current structure of the arguments suggests that they might be able to present their work in one well-structured and concise paper (or two) that really uses the data to substantiate the claims being made and demonstrates how they are meeting their Theory of Change which states: "The intended impact of PRiSE is to contribute towards the increased uptake and diversity of physics at higher education."**

We initially aimed to combine the current framework paper with the papers on impact and diversity. However, including sufficient information for clarity and to support our points made the paper exceedingly long, and awkwardly structured. Consequently, for

clarity, we split the manuscript into three papers. Each paper stands alone and does not rely upon the results of the others, however, we feel that reframing the papers to make their independence clearer and removing unnecessary cross references would assist readers. Recombining the three papers would not work as either a coherent (due to their different focuses) or concise (due to excessive length) paper. While we agree that doing so, on the surface, would seem to make sense, our experience highlights that it would not allow for a full exploration of all aspects of the programme and its evaluation to the level of rigour required by the journal.

**As a reader new to the program, and taking the abstract at face value, I found myself asking fundamental questions about the structure, resources, personnel and design of the program. This is touched on in brief in various parts of the paper but some challenges about the program structure are mentioned starting in line 500 that seem to warrant further comment, especially in the context of thinking about scalability of this program.**

We have tried to concisely discuss the structure, resources, personnel and design of the programme but appreciate there may be areas where more detail is required. We will re-examine the description to see where further elaboration is likely warranted.

**Other details like the core resources or research that undergird the program are mentioned in the latter part of the paper and seem a bit out of place. Perhaps some of the text could be instead captured in visuals or a diagram?**

We feel it is important for readers to gain some idea about the student activities involved in the current projects. We have tried to incorporate these elements in a concise manner, whilst also bearing in mind the range of backgrounds among the readership, though we would be happy to consider specific recommendations on visuals or diagrams and will re-visit this possibility.

**Overall, while the Theory of Change and surrounding literature review are helpful for framing the need and context of the PRiSE program, the body of the text and**

**data presented don't seem to directly align with or support the premise of the paper as articulated in the abstract and conclusions.**

Our premise in this paper is to detail the considerations made in developing the programme (in light of its aims), in particular detailing the structure, support, and resources offered by active researchers as part of PRiSE. Thus, the main bulk of the text and data presented are related to the provision and support offered as part of PRiSE and not the impact of the programme itself, as this is assessed in a different paper. However, in light of the reviewer's comments, we will try to more clearly articulate the premise of the paper in both the abstract and conclusions. Furthermore, as raised by other reviewers, we will aim to link back to the Theory of Change more throughout our discussion of the structure, support, and resources. Finally, upon reflection we feel that changing the title of this manuscript might also help in conveying its contents better, so we will alter this to "Developing a framework to bolster school students' aspirations through extended interaction with cutting-edge research: 'Physics Research in School Environments'".

**To concretely illustrate this, in the abstract the authors state "This illustrates that the model appears to provide highly positive experiences that are otherwise not accessible to schools and that the extraordinary level of support offered is deemed necessary with all elements appearing equally important. Researchers and public engagement professionals seem receptive to the PRiSE framework of schools engagement and it has started to spread to other institutions." While the authors present some data to demonstrate their programmatic success, for their most critical claims, they point the reader to a different paper (as above) and don't really touch on the focal point of their theory of change.**

Our intention was for the claims made in this paper only to pertain to the framework of the programme and how it is perceived by participants, rather than to the impact of the programme. The theory of change is presented to outline the aims of the programme and how that has influenced how we developed the programme's provision. This is similar to other published papers which introduce theories of change (e.g. Davenport

et al., 2020). We will edit the paper to make our premise for and focus in this paper clearer.

Davenport, C., Dele-Ajayi, O., Emembolu, I., Morton, R., Padwick, A., Portas, A., Sanderson, J., Shimwell, J., Stonehouse, J., Strachan, R., Wake, L., Wells, G., and Woodward, J.: A Theory of Change for Improving Children's Perceptions, Aspirations and Uptake of STEM Careers, Res. Sci. Educ., https://doi.org/10.1007/s11165-019-09909-6, 2020.

**Further, they mention on several occasions in the paper the "extraordinary level of support" needed and offered through this program by the researchers but do not elaborate on how this might be a barrier to the scalability of their program. It would be helpful to the readers if the authors were more explicit about how much time is required from researchers to support this type of programming, how researchers are recruited and rewarded/acknowledged for their participation and how the program itself is funded or supported, especially in light of the acknowledged barriers to sustaining engagement by researchers. These types of structural and programmatic details are key to seeing how the program supports their ToC and offers valuable insights for those seeking to recreate this type of 'research in schools' program.**

We have touched upon some of these elements within the manuscript, for example in discussing the different roles within the programme (lines 266-273). However, it is clear that more needs to be done. We will clarify in section 2.2 that the scalability we refer to mostly concerns the balance of reach and impact discussed here. We will also further explore aspects of scalability to other institutions, which is only briefly touched upon. We will also emphasise that many of these points will be highly dependent on specific structures and policies present within institutions (something which was briefly noted on lines 268-269).

**Building on the other reviewer's comments, I think there is ample opportunity to**

**streamline the text, and clarify the presentation to only those details most salient to communicating to the reader the design and implementation elements of the program, while being much more explicit about how the data they have collected demonstrate if/how (or not) the program 'meets' their Theory of Change.**

We agree that achieving an appropriate balance between being concise and providing sufficient detail is always a challenge. However, given that the other reviewers have lauded the detailed description of the various elements and that our aim with this paper is to provide sufficient detail required so that readers could replicate or create a similar programme, we hesitate to streamline or cut too much. While we hope that we have provided enough detail to exemplify relevant aspects of our Theory of Change (e.g. what the various inputs, activities and assumptions are), the exploration of potential impacts or outcomes are the subject of the accompanying paper.

**This is essential to demonstrate a) how the program is scalable and b) the documented value and impact and therefore, why it is a model that should be scaled to other schools/locations/programs.**

We agree with the reviewer that the paper would benefit from more exploration on scalability, which we will add. The value in terms of impact on participating students and teachers is explored elsewhere. While in some ways, it could be ideal to explore all of that in a single paper, this proved unfeasible.

**I would encourage the authors to significantly revise this manuscript and to think about how to present the details about how the program works and the data they have that indicates that the program is successful (and why) together in one paper.**

As noted, one single paper was not sufficient to fully explore all the aspects of the PRiSE programme and its evaluation to the level of rigour required by the journal.

**Based on the review criteria, this article falls short in demonstrating (in relation**

**to the other papers submitted for review at the same time) how each makes a unique and substantial contribution that warrants publication, and as currently written, this paper does not really provide sufficient evidence to support the interpretations and conclusions.**

The evidence and conclusions in this paper concern the experiences of participating students and teachers and their feedback on the level of support offered, as well as perceptions of researchers at other institutions about the potential scaling or spread of the framework and programme. We will try to ensure more clarity around the scope and aims of this particular paper, as well as noting that we do not aim to claim impact.

**I have no doubt that through some more careful writing, streamlining of the text and analysis of the data alongside the programmatic structure, that readers would see the substantial contribution being made through this program and its structure and the value it offers as a model that could be replicated elsewhere.**

**While there is work to do, I really do commend the authors on their thoughtful approach, clear investment in data collection and analysis, and for developing and iterating on a program that seeks to make a novel contribution for bringing research to schools. They certainly have invested an enormous amount of time and dedication to the PRiSE program and I really hope to see this work shared with the science communication and education communities.**

We thank the reviewer for their acknowledgement of the work that has gone into this programme and its evaluation and we hope that our changes will provide more clarity in the paper.

---

## Author Comment (AC6) · 6 Jan 2021

My co-authors and I have been carefully considering the editorial comments on our manuscript over the past several weeks. We certainly do think that we can improve the manuscript by making its purpose clearer and adding elements that reviewers felt were lacking. However, we feel that we would greatly benefit from some further clarifications on the comments in order to proceed to the best of our ability with these revisions, without which we foresee potentially even more rounds of lengthy revisions which I'm sure neither the editor or the reviewers want if can be avoided.

[Figure]

Firstly, I would like to make sure we on the same page as to the purpose of the manuscript – a point raised in the comments. We have revised the title and abstract as follows to hopefully make our intentions clearer.

**Title**: Developing a framework aimed at bolstering school students' aspirations through extended interaction with cutting-edge research: 'Physics Research in School Environments'

**Abstract**: Recent educational research has highlighted the distinction between "school" and "real" science, particularly acute in physics, and how this disparity may contribute to low and inequitable participation in science at higher education. 'Research in schools' initiatives which enable students to experience taking part in cutting-edge research within their school over several months are a relatively new form of independent research project which have emerged that may have some role to play within this. However, at present the different models of provision within 'research in schools' projects and whether they are sufficient remain unclear. This paper explores the development of a scalable framework for 6-month-long 'research in schools' projects which has now been running for 6 years. By constructing a Theory of Change, we break down the aims of this extended programme into a series of realistic intended outcomes for diverse groups of participating 14–18 year old students, their parents/carers, and teachers and wider school environment. Based on this Theory of Change, we discuss the considerations made within the developed framework, in particular detailing the structure, support, and resources offered by active researchers. The framework is evaluated through feedback from participating students and teachers. This illustrates that the model appears to provide highly positive experiences that are otherwise not accessible to schools and that the level of support offered is deemed necessary with all elements appearing equally important. Researchers and public engagement professionals seem receptive to the framework and we suggest that it could be adopted at other institutions applied to their own areas of scientific research, something which has already started to occur.

The material included in the submitted manuscript to align to this focus was based on discussions not only amongst co-authors, but those acknowledged at the end of the manuscript as well as editors and reviewers of a previously submitted paper, which recommended distributing the full discussion of PRiSE across several papers and outlining specifically what should be included within each of them. Therefore, we really need more guidance as to what the editor thinks we should cut. On our own re-reading, while there are certainly some sentences or paragraphs which we think could be removed, largely our opinion is that all the sections need to remain in some form. We briefly detail why

1. Introduction: Sets the background context

2. PRiSE framework

   (a) Aims: A programme must be developed with clear aims in mind and we present these through a Theory of Change

   (b) Reach: We will reframe this section to be more explicitly about scalability, as was intended, moving it later in the manuscript.

   (c) Approach: We will split this up into more explicit subsections and ensure we tie these back to the Theory of Change, but the content explores ethical considerations and roles of teachers and researchers.

   (d) Structure: This details the structure, interventions and resources developed for this 6-month-long programme with each element only getting a paragraph. Even with a graphical representation (which we have made and attached) we feel all these elements still need some discussion.

(e) Current projects: We could summarise these in a table and relegate specifics about current projects to supplementary material, as we still think they are useful details to practitioners.

3. Methods: Outlines the surveys and analysis techniques used to evaluate the developed framework

4. Feedback from participants: Results on experience and support within our framework.

5. Feedback from the university sector: Perceptions from other researchers and practitioners as to scalability.

6. Conclusions

Therefore, we would really appreciate further input.

Finally, the editor comments say we should incorporate the most relevant reviewer comments. To our mind these concern:

- Detailing how each element of the framework aligns with our Theory of Change

- More explicit discussion of the scalability of the framework

We will endeavour to make these changes as they will improve the manuscript. However, we are concerned with the contradictory nature of RC5 to RCs 1-4 and thus how to respond accordingly. More direction from the editor would therefore be most helpful.

**Activity Stages:**

| Jun | Jul | Aug | Sep | Oct | Nov | Dec | Jan | Feb | Mar |

Prescribed work

Independent project

Writing up

**Interventions:**

**Assignment**

**Kick-off**
(1.5h on campus)

**Webinars**
(optional, 1h monthly)

**School visits**
(1-2 per school, 1h each)

**Ad hoc support**

**Comments**

**Conference**
(2.5h on campus)

Outreach Officer
(Programme management;
1 per department)

Project Lead
(Figurehead; 1 per project)

Researcher
(Support & mentor students/
teachers; ≥1 per project)

Teacher
(Encourage students & communicate
with university; ≥1 per school)

Student
(No limit per school,
work in groups of ~5)

Parent/carer

Advertise & assign

Apply

Organise event, brief students & teachers

Science talk

Facilitate workshop

Recruit students

Communicate to teachers

Respond and give guidance

Ask questions

Ask questions, show work

Communicate to teachers

Respond and give guidance

Arrange, ask questions

Ask questions, show work

Respond to issues with guidance

Communicate issues to university

Report issues to teacher

Provide comments

Co-ordinate draft submissions

Organise & run event

Interact with students & judge outputs

Interact with students & judge outputs

Co-ordinate submissions, support students

Present posters or talks

Support students

**Resources:**

Project poster/flyer

Student/ teacher project guide

Project webpages

General `how to' guides

**Fig. 1.** Framework diagram

---

## Editor Comment (EC1) · John K. Hillier (Editor) · 8 Jan 2021

Dear Martin,

Thank you for your comment. I have looked at your comments, the submitted paper, and the two companion papers you have submitted, alongside the reviewer comments. Your proposal at the moment still reads, to me at least, as a broad and sweeping outline and discussion of PRiSE, rather than a clearly focussed piece. I still find it hard to discern the specific academic purpose of the paper, which will allow you to simplify and clarify it. To this end, I have taken the liberty of attempting to re-write the abstract and make a few suggestions. I realise that it is difficult to step back, view with fresh eyes, and substantially modify a paper with a substantial history, but I encourage you to attempt this.

The abstract I suggest is shorter, and gives you space to firstly describe PRiSE, then after this set-up get onto the research that is the core of the paper. Currently the introductory description of PRiSE is 18 pages, and all of the research (methods/results etc ...) is 10 pages. Although this paper can be to some extent a vehicle for a description of PRiSE, this balance may explain some of the reviewer reactions. The clear description of the paper's content I suggest may mitigate this partially, but some re-balancing is probably also necessary, using appendices if you must. In terms of what should be cut, the 10 pages of research seems to be a reasonable length.

> **Title:** Evaluating participants' experience of an initiative (PRiSE) to inspire school students to continue studying physics by extended interaction with cutting-edge research.

> **Abstract:** Physics in schools is distinctly different from university, research-level work, which may hinder participation in science at higher education. Initiatives wherein students engage in independent research linked to cutting-edge research within their school over several months may mitigate this. However, how this is best done remains unclear. This paper evaluates the PRiSE initiative through participants' experience of the scalable framework used. First, the PRiSE initiative and the theory of change used to break down its aims into a series of realistic intended outcomes for 14–18-year-old students are described. Then, the framework used is evaluated using survey data from participating students, teachers, and university collaborators. Overall, PRiSE appears to provide highly positive experiences that schools cannot provide internally, and the intensive support offered is deemed necessary with all elements appearing equally important. We suggest that the framework could be adopted at other institutions and applied to their own areas of scientific research, something which has already started to occur.

Please find below a non-exhaustive list of comments that I hope might help. Please also respond to all of the reviewers' comments, although because you are seeking to focus and shorten the piece this may be given as a valid reason for not acting upon some of them.

- "Developing" in the title. Developing, implies a narrative description of the process of development, which is difficult to reconcile with a research article.
- "PRiSE is a positive experience", but for a research article there is a need to focus on whether this is better or worse than other frameworks/initiatives.
- Scalability, if emphasized, needs to be evaluated. If there is no empirical evidence to assess this, I suggest you downplay it; including, for example a paragraph describing how the framework might be scaled up.
- Please seek to state your work concisely e.g. the abstract is now 160 words down from 264. You may be assisted in making the paper concise once the purpose of the paper is clearly identified. An illustration of this is Section 2.5 in version 1 of the manuscript ("Current Projects"). This probably needs no more than a relatively short paragraph and a single line for each project. Please ask yourselves for each section: How is this critical to either (i) a

clear concise description of the framework or (ii) its evaluation. Other material should be removed please, although appendices can be used if needed. I also suggest keeping descriptions of parts of the framework that are uncontroversial or not evaluated short, allowing some more space for the elements of interest to the research into participants' experience you present.

- I changed the text to "schools cannot provide internally" as the experiences could be possible through other frameworks / initiatives.
- I removed inverted commas and jargon from the first sentences of the abstract for clarity of communication.

---

## Author Response (AR1)

**Response to reviewers**

Transforming school students' aspirations into destinations through extended interaction with cutting-edge research: `Physics Research in School Environments'
M.O. Archer et al.

We thank the editor and the reviewers for their comments. We have revised the manuscript in response to these, which we detail here. Line numbers refer to the tracked changes version of the manuscript.

**EC1**

**Thank you for your comment. I have looked at your comments, the submitted paper, and the two companion papers you have submitted, alongside the reviewer comments. Your proposal at the moment still reads, to me at least, as a broad and sweeping outline and discussion of PRiSE, rather than a clearly focussed piece. I still find it hard to discern the specific academic purpose of the paper, which will allow you to simplify and clarify it. To this end, I have taken the liberty of attempting to re-write the abstract and make a few suggestions. I realise that it is difficult to step back, view with fresh eyes, and substantially modify a paper with a substantial history, but I encourage you to attempt this.**

**The abstract I suggest is shorter, and gives you space to firstly describe PRiSE, then after this set-up get onto the research that is the core of the paper. Currently the introductory description of PRiSE is 18 pages, and all of the research (methods/results etc ...) is 10 pages. Although this paper can be to some extent a vehicle for a description of PRiSE, this balance may explain some of the reviewer reactions. The clear description of the paper's content I suggest may mitigate this partially, but some re-balancing is probably also necessary, using appendices if you must. In terms of what should be cut, the 10 pages of research seems to be a reasonable length.**

**Title: Evaluating participants' experience of an initiative (PRiSE) to inspire school students to continue studying physics by extended interaction with cutting-edge research.**

**Abstract: Physics in schools is distinctly different from university, research-level work, which may hinder participation in science at higher education. Initiatives wherein students engage in independent research linked to cutting-edge research within their school over several months may mitigate this. However, how this is best done remains unclear. This paper evaluates the PRiSE initiative through participants' experience of the scalable framework used. First, the PRiSE initiative and the theory of change used to break down its aims into a series of realistic intended outcomes for 14–18-year-old students are described. Then, the framework used is evaluated using survey data from participating students, teachers, and university collaborators. Overall, PRiSE appears to provide highly positive experiences that schools cannot provide internally, and the intensive support offered is deemed necessary with all elements appearing equally important. We suggest that the framework could be adopted at other institutions and applied to their own areas of scientific research, something which has already started to occur.**

We now understand the editor's position much more clearly following these helpful comments. We have largely used the editor's suggested title and abstract and have dramatically reduced the length of the description of PRiSE in the article, taking the editor's broad comments in mind.

**Please find below a non-exhaustive list of comments that I hope might help. Please also respond to all of the reviewers' comments, although because you are seeking to focus and shorten the piece this may be given as a valid reason for not acting upon some of them.**

**• "Developing" in the title. Developing, implies a narrative description of the process of development, which is difficult to reconcile with a research article.**

We have used wording more aligned with the editor's suggestion in the title.

**• "PRiSE is a positive experience", but for a research article there is a need to focus on whether this is better or worse than other frameworks/initiatives.**

We now emphasise that our evaluation shows PRiSE gives a more positive experience to participants than typical schools engagement programmes, as demonstrated by comparing to benchmark data.

**• Scalability, if emphasized, needs to be evaluated. If there is no empirical evidence to assess this, I suggest you downplay it; including, for example a paragraph describing how the framework might be scaled up.**

Scalability has been downplayed more than in the previous version. We now provide a brief discussion in section 2.3, demonstrating how the PRiSE model through its efficient use of researchers' time has allowed more schools to be involved per institution than other formats. Section 5 also presents data from independent researchers and engagement professionals, with one of the resultant overall themes being that approaches like PRiSE are deemed achievable by them.

**• Please seek to state your work concisely e.g. the abstract is now 160 words down from 264. You may be assisted in making the paper concise once the purpose of the paper is clearly identified. An illustration of this is Section 2.5 in version 1 of the manuscript ("Current Projects"). This probably needs no more than a relatively short paragraph and a single line for each project. Please ask yourselves for each section: How is this critical to either (i) a clear concise description of the framework or (ii) its evaluation. Other material should be removed please, although appendices can be used if needed. I also suggest keeping descriptions of parts of the framework that are uncontroversial or not evaluated short, allowing some more space for the elements of interest to the research into participants' experience you present.**

We have removed all parts that we feel are not critical based on the editor's comments.

**• I changed the text to "schools cannot provide internally" as the experiences could be possible through other frameworks / initiatives.**

We have taken this into account in the abstract.

**• I removed inverted commas and jargon from the first sentences of the abstract for clarity of communication.**

We have inverted commas and jargon from the abstract.

**Editor comments to author**

**Three reviewers found the work interesting, well presented, and with a good grasp of best practice. One reviewer [RC5], however, suggested some major revisions. These relate to the presentation and focus of the material, rather than the quality and detail of the content. Please give these concerns serious attention. Overall, I think it has the potential to be an excellent paper, and very much encourage you to undertake the revisions necessary.**

**I note that you have been in touch with the editorial team of GC more generally about the suite of papers submitted about the PRiSE project. Thus, an overarching consideration for the paper is an understanding that you will sharpen the focus of each paper.**

We clarify that the executive editor contacted us in the first instance and we responded.

**In terms of an overview on how to approach the revision, given my initial reaction (see below) I find it hard to express it better than in RC5**

**"Building on the other reviewer's comments, I think there is ample opportunity to streamline the text, and clarify the presentation to only those details most salient to communicating to the reader the design and implementation elements of the program, while being much more explicit about how the data they have collected demonstrate if/how (or not) the program 'meets' their Theory of Change. This is essential to demonstrate a) how the program is scalable and b) the documented value and impact and therefore, why it is a model that should be scaled to other schools/locations/programs."**

**In my words: The purpose of the paper should be clearly laid out, and it should be readily apparent why material retained is directly related to this - if not, remove it. Illustratively, in the first line of the abstract you say "We introduce a scalable framework .... " Why do you introduce it? In the second sentence you say what PRiSE's aim is, but you do not clearly state what the aim of this paper is in the abstract. Clarity here might help both writer and reader.**

Following EC1 we now better understand the editor's position and have throughout attempted to clarify the purpose of the paper and why its contents are included. Much material has been removed in the revision to keep the description of PRiSE concise.

**• Creating a paper that is readable, focussed and concise, incorporating the reviewers'comments that are most relevant.**

We have attempted to refocus the content and reduce its length, while also addressing the reviewer comments relevant to the purpose of the paper.

**Focus on how the distinct added value and purpose of this paper, as opposed to the other papers.**

All cross-references to other papers have been removed from the body of the article.

**Attempt to briefly consider international initiatives.**

We have briefly added to the introduction some relevant international initiatives (lines 67-75) as well as expanding discussion of the HiSPARC programme to include the other countries in which it runs (lines 93-95).

**1. Please attempt to make the manuscript more concise. It seems quite long for thematerial presented.**

We have reduced the length of the manuscript.

**2. I am aware of, and have participated in the Nuffield STEM initiative that has been running for at least 10 years (https://www.nuffieldfoundation.org/contact). By linking school students to active research, placing them in universities it performs a similar role, and includes physics within its remit (i.e. one of my students investigated an element of geophysics, winning an award).**

While this is a different format to `research in schools', we now highlight on lines 67-71 some initiatives that make use of dedicated out-of-school events including Nuffield placements.

**3. The paper is currently very UK-centric for an international journal. Please make an attempt to identify and acknowledge other initiatives globally. It is difficult to believe that these do not exist, but if they do not, then please argue this case explicitly.**

**• RISE programme at Stanford - summer internship programme.**

**https://oso.stanford.edu/programs/disciplines/20-physics**

**• https://www.sas.upenn.edu/summer/programs/high-school/experimentalphysics**

**• 'ANU extension' https://physics.anu.edu.au/engage/outreach/**

We have included relevant examples of international programmes on lines 67-75. Our discussion of the HiSPARC programme now also includes the other countries in which it runs (lines 93-95).

**4. If PRiSE is scalable, can you provide a simplified diagram that others could use to set up similar programmes, perhaps in other countries or other scientific fields? If it is purely physics (excluding physics in related disciplines) and only in the UK, state limitations on scope at the start. [see RC3 point 3]**

We have created a figure (Figure 2 in the tracked changes version) to summarise the framework.

**L106 - "complete lack"? It would be good to see a review of other papers investigating schemes that use research in school as a method; these might be academic papers, but internal evaluations of these schemes or grey-literature they have published would be beneficial here. Please add a paragraph.**

We have undertaken a comprehensive literature review of both academic and published grey-literature surrounding both independent research projects and `research in schools' specifically. This has all been included in the introduction in the preceding paragraphs. Unfortunately very little material surrounding `research in schools' style initiatives have been made public, highlighting the need for this paper.

**L575-580 ..... discussion of causes of this expected later.**

The qualitative research starting on line 691 investigates the causes of these positive quantitative results. We have now made this clearer.

**L593 ... how were the categories / themes defined? Add reference for analysis method please.**

This was outlined in our Methods section on lines 654-660.

**L806 - some formatting issues with references.**

So readers do not confuse papers by L. Archer with the first author we have disambiguated the two by including initials. This has been checked with Copernicus staff.

**RC1**

**The article submitted about the PRiSE project is very interesting. The quality of the presentation is excellent ... the implementation and results of the project are clearly presented. Nevertheless, I would have liked to have seen more details for the teachers involved in the project. What strategy is actually implemented for teacher training during the PRiSE project?**

**line 172 > 'Therefore, opportunities for teachers' development are integrated within the programme rather than being a separate offering to schools'**

We have added a paragraph on lines 266-271 about teacher development.

**RC3**

**Many thanks to authors for all of their efforts in putting together this interesting piece of research. I am sure that many of those working in Outreach and Public Engagement in HE could use some elements of this article for their own benefit. That said there are few elements I ask to authors to review. These comments should be seen as constructive and should really enhance the current structure of the article**

**1. The Theory of Change (ToC) presented reads well and it follows very closely the Theory of Change published by Davenport et al 2020., in terms of identified audiences, causal paths and terminology. I recommend a more explicit acknowledgement by the authors to Davenport et al 2020. Where I think there is room for improvement is to explain or summarise and assumptions and barriers that often accompany a ToC, as throughout the paper some of these emerge (e.g - researchers or institutional buy-in a barrier to your ToC ). Finally the ToC description needs a bit more details. For instance what is the meaning of the different shades of the same colour**

Upon the editor's request to refocus the article, we have removed the Theory of Change.

**2. The framework as well described by the authors, feels disconnected from the ToC and more references throughout the text should be made to the ToC especially in section 2.3 and 2.4 For example line 230 to 245 , removal of barriers, involvement of teachers etc, really highlight that these are aspects of your ToC.**

Upon the editor's request to refocus the article, we have removed the Theory of Change.

**3. Even though the PRISE framework has been presented as scalable, what are the lessons learnt by the authors? what are the recommendations to other practitioners in the field? Adding a few paragraphs in the conclusions, or even some bullet points, would address not only the scalability of PRISE but the transferability of PRISE to another subject or institution (e.g. - produce a detailed guide for students and teachers, etc)**

We have added sentences highlighting the recommendations to practitioners based on the results of the evaluation. These can be found on lines 908-915.

**RC4**

**General comments**

**This paper makes a valuable contribution to the available literature on undertaking projects that involve school students in research. It is generally well-constructed and well written, leading the reader through the premise, structure and success of the programme. The detailed exposition of the workings of PRiSE is especially welcome as it facilitates the successful replication of such a programme without a duplication of the evidently extensive effort and multiple trials that have been required to bring the programme to its current level.**

**There are some points in the paper (as will be addressed in the specific comments below) that would benefit from further consideration; however, these comments are mostly fairly minor, and**

**are noted in a spirit of bringing the level of every part of the paper to the high standard it exhibits overall.**

**The authors give a thorough depiction of the landscape in which this work sits, taking care to give details of other similar projects distinct from PRiSE. Nevertheless, as noted below, these other projects are not always considered in a positive light. It might be wise not to be over-critical at the risk of sounding petty rather than constructive. However, proper credit is given where appropriate, both to work outside this project and to the researchers and other staff involved in PRiSE, which was heartening to see.**

We have adjusted the tone slightly in places when discussing other projects to mitigate any perceived negativity. The points raised summarise the information available about these projects, highlighting the need for more publications detailing the provision within this area, and also to be able to compare/contrast to PRiSE's approach.

**The title seems fair, although there is an emphasis on 'destinations' that is less apparent within the body of the paper. Although this is mentioned within the section on the Theory of Change, there seems to be little further discussion or evidence of the destinations of students that take part in PRiSE. Nevertheless, the abstract provides a concise, complete and clear summary of the contents of the paper.**

Upon the editor's request to refocus the article, we have changed the title.

**The language is largely fluent and precise. On occasion, some of the sentence structures are a little hard to follow on a first reading. In particular, there is substantial use of possessive apostrophes that on occasion impede initial comprehension. It may be worth reconsidering some of these to aid the flow of the text (as opposed to the text's flow).**

We have attempted to improve the language throughout.

**The paper is well-referenced throughout, with many recent publications cited, demonstrating a laudable grasp of current best practice and educational research. This is to be highly commended.**

**Specific comments**

**The specific comments are given with line references relating to the pre-print (pdf) of the paper.**

**- Line 9: 'with all elements appearing equally important.' – it would be useful (perhaps later in the article) to have a simple list of all the elements that are being encompassed by this phrase.**

The added figure (Figure 2 in tracked changes version) attempts to make the elements of the PRiSE programme clearer.

**- Line 66 on: The discussions of other similar projects, while not obviously straying from factual, nevertheless read as ungenerous. E.g. line 73: 'other memberships are seemingly justified to ensure that schools make a commitment to working with the university'; line 88: 'While some researchers/academics have designed or consulted on some IRIS projects, they appear in general to have little involvement supporting students or teachers'. This could perhaps be construed as criticism of the other projects (with the aim of elevating PRiSE) which may or may not be considered constructive at this juncture.**

As mentioned earlier, we have adjusted the tone of these statements. See lines 101 and 117-118.

**- I understand that the Theory of Change as presented here is discussed in more detail in another publication. Nevertheless, I would query a couple of aspects that are here presented without substantial examination (though I recognise this review comment may not be the best home for this remark and the authors may feel that no response or alteration is merited.) ** Figure 1: The implication that 'Know other people interested in physics' leads to 'See themselves as equals in physics to those from different backgrounds'. I don't know that this follows. I think you can quite easily know other people interested in physics and *not* see yourself as "equal in physics" to those other people. ** Line 163: 'By interacting first-hand with "real physics" through the projects and working with active researchers, students (especially those from underrepresented groups) should feel included'. I think it is perfectly possible to do physics research and yet feel excluded. I am not convinced that under-represented students will automatically feel included, simply by virtue of doing "real physics", especially if they do not recognise themselves in the active researchers they are connected with or are a minority in the group taking on a project in their school.**

Upon the editor's request to refocus the article, we have removed the Theory of Change.

**- Line 201 -205. How do the other IRIS physics projects compare here? As it stands, it reads like a cherry-picked list of worst performers, highlighted to make PRiSE look good. If further data on the numbers of researchers / schools is not available for other IRIS projects, then this is worth noting here to avoid this impression.**

We now note that information on the other IRIS projects has not been made available (line 596).

**- Line 196: It becomes apparent here that some schools have taken part but then dropped out. It might be worth pointing this out explicitly, and possibly signposting the later short discussion of this (e.g. around line 806)**

This is now explicitly pointed out on lines 249 and 587-590.

**- Line 240 – 242: teachers decide who to offer the PRiSE projects to. Do you have any thoughts on how successful teachers are at selecting students who excel on the projects?**

Unfortunately, we don't have any specific information on how teachers go about selecting students, which we now raise in the manuscript on lines 258-261.

**- Line 248: 'We allow teachers to determine how best to integrate the projects within their school, though provide advice on this.' From the perspective of an outreach practitioner hoping to replicate the success of PRiSE, it would be interesting and useful to see this advice – perhaps included in an appendix?**

As this had been done informally, unfortunately we cannot include this material.

**- Line 286 on: How much drop-off do you typically see between teachers applying for projects and then not taking up an offer come the new academic year?**

There is a 33±5% drop-off between application/assignment and the initial kick-off meeting in the new academic year. Retention within the programme is outside the scope of this paper and is addressed in another publication.

**- Line 309: 'though this latter approach often proves unsuccessful' – thank you for including this kind of helpful detail**

**- Line 463: What is 'the UK coding agenda'? This phrase needs further explication and / or a reference**

Upon the editor's request to refocus the article this is no longer referenced.

**- Line 534: Feedback from the university sector. This is a bit confusing – it's a little unclear what the university sector is being asked or why, and how that connects with the previous discussion of participant feedback. Although there will be further detail given later, it might be worth clarifying some of it at this stage. Maybe it's simply the mention of "the workshop" (line 535) without context that is disconcerting.**

The content of the workshop is now clarified on lines 631-633 and 851-852.

**- Line 589: I enjoyed the inclusion of the negative words in the word cloud, and appreciated that they were highlighted here.**

It is important that the entirety of the collected data are presented and discussed in a balanced and appropriate way, which we have aimed to do throughout.

**- Line 791: 'The ethos behind PRiSE is to transform current scientific research methods' This could be read as though you are trying to alter the way the scientists undertake their research. Perhaps consider re-phrasing this, if that is not your intention.**

This sentence is no longer included.

**Technical corrections**

**- Line 143: 'with standard one-off (or even short-series of) intervention(s)' – this doesn't quite read right to me**

Upon the editor's request to refocus the article this is no longer included.

**- Line 169: 'Experience from physics outreach officers . . . have shown' – grammar error. Should either be 'Experiences . . . have shown' or 'Experience . . . has shown'**

Upon the editor's request to refocus the article this is no longer included.

**- Line 174: 'the impacts of PRiSE can be felt much wider' – grammar. Suggest 'can be felt much more widely'.**

Upon the editor's request to refocus the article this is no longer included.

**- Line 183: 'another major influence on young people's aspirations are family' – grammar error. Should be 'another major influence . . . is family'.**

Upon the editor's request to refocus the article this is no longer included.

**- Line 213: 'One might think it is feasible that students' work on PRiSE projects contribute to novel research.' - Grammar: contributes - This sentence is generally hard to follow – consider revising**

This has been rephrased, see lines 298-299.

**- Line 423: 'organic semiconducters' – typo: semiconductors**

Upon the editor's request to refocus the article this is no longer included.

**- Line 502: 'and responsibilities have remained largely been falling to only a few people per PRiSE project' – grammar. Remove 'remained'?**

Upon the editor's request to refocus the article this is no longer included.

**- Line 803: 'and have relished the challenge of working differently to in their regular school experience' – grammar. Remove 'in'?**

This correction has been made, see line 908.

**RC5**

**Thank you to the authors for working to summarize their program and research as part of the PRiSE program. It is clear that this group has taken the task of creating and evaluating their educational program seriously and I commend them on identifying many different facets of the program to document and share with the broader community. The paper is generally well presented but as a reader first learning about this program I have some major revisions to suggest.**

**Major Considerations for Revision: While I see the clear value and need to share this work with the broader community, especially given the authors' goal to "introduce a scalable framework for protracted research-based engagement with schools", I have some questions about how this and the other papers submitted simultaneously in review in Geoscience Communication about the same program differ from one another and how they each meet the journal requirement of making "a substantial contribution to scientific progress within the scope of Geoscience Communication (substantial new concepts, ideas, methods, or data)"**

The three papers concern separate areas:

- This paper: A process evaluation of participants' experience within our provision framework.
- Impact evaluation exploring the actual benefits for students and teachers that might have resulted from the programme.
- Audience evaluation assessing the diversity, accessibility and equity within the programme for the schools we work with.

Each paper contains clear and separate conclusions. Other published papers in this field by other authors, for example the various papers cited on IRIS's programme, also split the content and focus up in similar ways thus we feel this is appropriate. Upon discussion with the executive editorial team at Geoscience Communication they are satisfied with our response that three papers are required.

**Further, based on the references cited in this manuscript, the lead author also has another paper in review at another journal that seems to align with a similar premise being presented here.**

Upon the editor's request to refocus the article this is no longer referenced.

**As a reader (who hasn't read all of these manuscripts to know exactly how they differ), I'm left wondering why someone would need to read four papers about a program to understand the structure and impact.**

The programme is significantly more extended, consisting of several different parts, than typical one-off engagement approaches published in this journal and others, thus it is not surprising that a full exploration of its structure and impact is lengthy.

**While I fully appreciate the appropriateness of evaluating and interpreting results from a program like this in multiple ways, the current structure of the arguments suggests that they might be able**

**to present their work in one well-structured and concise paper (or two) that really uses the data to substantiate the claims being made and demonstrates how they are meeting their Theory of Change which states: "The intended impact of PRiSE is to contribute towards the increased uptake and diversity of physics at higher education."**

We initially aimed to combine the current framework paper with the papers on impact and diversity. However, including sufficient information for clarity and to support our points made the paper exceedingly long, and awkwardly structured. Consequently, for clarity, we split the manuscript into three papers. Each paper stands alone and does not rely upon the results of the others. The papers have been reframed to make their independence clearer and unnecessary cross references have been removed. Upon the editor's request to refocus the article, we have removed the Theory of Change. Impact is not within the scope of this paper.

**As a reader new to the program, and taking the abstract at face value, I found myself asking fundamental questions about the structure, resources, personnel and design of the program. This is touched on in brief in various parts of the paper but some challenges about the program structure are mentioned starting in line 500 that seem to warrant further comment, especially in the context of thinking about scalability of this program.**

We have attempted to clarify the structure, resources, personnel and design of the program of the programme throughout the revised section 2. Challenges are also more explicitly highlighted.

**Other details like the core resources or research that undergird the program are mentioned in the latter part of the paper and seem a bit out of place. Perhaps some of the text could be instead captured in visuals or a diagram?**

We have summarised the specific projects in Table 1 and added a figure (Figure 2 in tracked changes) to outline the framework.

**Overall, while the Theory of Change and surrounding literature review are helpful for framing the need and context of the PRiSE program, the body of the text and data presented don't seem to directly align with or support the premise of the paper as articulated in the abstract and conclusions.**

Upon the editor's request to refocus the article, we have removed the Theory of Change.

**To concretely illustrate this, in the abstract the authors state "This illustrates that the model appears to provide highly positive experiences that are otherwise not accessible to schools and that the extraordinary level of support offered is deemed necessary with all elements appearing equally important. Researchers and public engagement professionals seem receptive to the PRiSE framework of schools engagement and it has started to spread to other institutions." While the authors present some data to demonstrate their programmatic success, for their most critical claims, they point the reader to a different paper (as above) and don't really touch on the focal point of their theory of change.**

Upon the editor's request to refocus the article, we have removed the Theory of Change.

**Further, they mention on several occasions in the paper the "extraordinary level of support" needed and offered through this program by the researchers but do not elaborate on how this might be a barrier to the scalability of their program. It would be helpful to the readers if the authors were more explicit about how much time is required from researchers to support this type of programming, how researchers are recruited and rewarded/acknowledged for their**

**participation and how the program itself is funded or supported, especially in light of the acknowledged barriers to sustaining engagement by researchers. These types of structural and programmatic details are key to seeing how the program supports their ToC and offers valuable insights for those seeking to recreate this type of 'research in schools' program.**

The added figure (Figure 2 in tracked changes) now explicitly highlights time commitments of each stage of the framework. Scalability is now briefly discussed in section 2.3 and acknowledges the barriers to sustained engagement by researchers. Views from independent researchers are presented in section 5.

**Building on the other reviewer's comments, I think there is ample opportunity to streamline the text, and clarify the presentation to only those details most salient to communicating to the reader the design and implementation elements of the program, while being much more explicit about how the data they have collected demonstrate if/how (or not) the program 'meets' their Theory of Change.**

Upon the editor's request to refocus the article, we have removed the Theory of Change from the article and streamlined the text.

**This is essential to demonstrate a) how the program is scalable and b) the documented value and impact and therefore, why it is a model that should be scaled to other schools/locations/programs.**

Scalability is now briefly discussed in section 2.3. The impact of the programme is explored in another publication.

**I would encourage the authors to significantly revise this manuscript and to think about how to present the details about how the program works and the data they have that indicates that the program is successful (and why) together in one paper.**

As noted, one single paper was not sufficient to fully explore all the aspects of the PRiSE programme and its evaluation to the level of rigour required by the journal.

**Based on the review criteria, this article falls short in demonstrating (in relation to the other papers submitted for review at the same time) how each makes a unique and substantial contribution that warrants publication, and as currently written, this paper does not really provide sufficient evidence to support the interpretations and conclusions.**

The evidence and conclusions in this paper concern the experiences of participating students and teachers and their feedback on the level of support offered, as well as perceptions of researchers at other institutions.

**I have no doubt that through some more careful writing, streamlining of the text and analysis of the data alongside the programmatic structure, that readers would see the substantial contribution being made through this program and its structure and the value it offers as a model that could be replicated elsewhere.**

**While there is work to do, I really do commend the authors on their thoughtful approach, clear investment in data collection and analysis, and for developing and iterating on a program that seeks to make a novel contribution for bringing research to schools. They certainly have invested an enormous amount of time and dedication to the PRiSE program and I really hope to see this work shared with the science communication and education communities.**

---

## Referee Report (RR1)

**Transforming school students' aspirations into destinations through extended interaction with cutting-edge research: 'Physics Research in School Environments'**

Archer et al.

Editor's Review of R1

Before considering any previous comments or responses, I reviewed the R1 manuscript with the aim of seeing it afresh as a *Geoscience Communications* reader might.

The paper evaluates the PRiSE framework, via the experiences of students and teachers, primarily based upon survey data. It finds that the extended timeframe of contact and support is important, and that the teachers and students see value in all elements of the support offered within the framework.

The Introduction is long, but excellently written and draws the reader into precisely what makes PRiSE distinctive, and thus why an evaluation of PRiSE is a useful contribution to the scientific community as a whole (students, teachers and academics). Section 2 is again sizeable, but with the evaluative purpose of the paper clearly stated in the Abstract, it is necessary and useful to set the scene. The data collection and analysis are clearly and reproducibly set out, statistically robust, with useful figures to summarize key results (i.e. 3-5). The thematic analysis is very detailed, long, but not quite disproportionately so.  Together, the work allows the final section to consist of unambiguously well-supported conclusions, which contain clear and important messages.

Major points

1.    From a detailed read of the new manuscript, then the authors' responses to all previous comments, it is clear the authors have done an excellent job of responding and have made changes that do everything asked of them. This is now an impressively well-constructed and presented manuscript.

Minor points/comments

L4 - Please satisfy yourself that 'provision framework' is a thing. Sounds a little odd, but I can't immediately suggest a change.
L8 - comma before 'with'?
L47 - Nice that international experience is recognised.
L50 - 'This review ....' ?
L52 - This chimes with my experience of supervising high-school students on extended projects.
L55-65 - Good to place PRiSE in the context of other related schemes so that it's benefits can be clearly identified.
L67 - rather than '.... typically one ....' <5 perhaps might be fairer than 1, from my memory of Nuffield 2-3 was quite common although 1 was not unusual. '1 to 3'? It doesn't harm your point.
L135 - Use of supplementary material to avoid lengthy digressions from the stated point of the paper works.
L144-155 - Please consider placing this in a sub-section 2.1.1 highlighting ethics e.g. 'Ethical considerations'. This is something of importance to GC's ethos. If you choose to, renumber following sections i.e. 2.1.2 for Roles of Teachers etc ....
L195 - Motivation of academics is a useful and important aspect to consider. Well put.
L198 - I agree it's definitely 'can' not 'is'. This is not as developed as it could be.  Hillier et al (2019) 'Demystifying Academics' in *GC* (https://gc.copernicus.org/articles/2/1/2019/) includes an analysis of how impact is included in academics promotion criteria in the context of how they/we are motivated.
L557-559 - Your choice, but a somewhat negative tone. e.g. L559 instead of 'no data' you could just state that 'We have also gathered data on the impact of the PRiSE initiative, which is assessed elsewhere (Archer & Witt, 2020)'

---

## Editor Decision (ED1)

Three reviewers found the work interesting, well presented, and with a good grasp of best practice. One reviewer [RC5], however, suggested some major revisions. These relate to the presentation and focus of the material, rather than the quality and detail of the content. Please give these concerns serious attention. Overall, I think it has the potential to be an excellent paper, and very much encourage you to undertake the revisions necessary.

I note that you have been in touch with the editorial team of *GC* more generally about the suite of papers submitted about the PRiSE project. Thus, an overarching consideration for the paper is an understanding that you will sharpen the focus of each paper.

In terms of an overview on how to approach the revision, given my initial reaction (see below) I find it hard to express it better than in RC5

"Building on the other reviewer's comments, I think there is ample opportunity to streamline the text, and clarify the presentation to only those details most salient to communicating to the reader the design and implementation elements of the program, while being much more explicit about how the data they have collected demonstrate if/how (or not) the program 'meets' their Theory of Change. This is essential to demonstrate a) how the program is scalable and b) the documented value and impact and therefore, why it is a model that should be scaled to other schools/locations/programs."

In my words: The purpose of the paper should be clearly laid out, and it should be readily apparent why material retained is directly related to this - if not, remove it. Illustratively, in the first line of the abstract you say "We introduce a scalable framework .... " Why do you introduce it? In the second sentence you say what PRiSE's aim is, but you do not clearly state what the aim of this paper is in the abstract. Clarity here might help both writer and reader.

Please:-

- Creating a paper that is readable, focussed and concise, incorporating the reviewers' comments that are most relevant.
- Focus on how the distinct added value and purpose of *this* paper, as opposed to the other papers.
- Attempt to briefly consider international initiatives.

Importantly, in your revision please provide a manuscript with (simplified) tracked changes, indexed to a point-by-point response to all of the reviewers' comments. Please consider both adding Word 'Comments' in the text to refer back to comment numbers, and referring to line numbers (revised manuscript) in your point-by-point response. This will allow me to assess the revision.

All the best,

John

Editorial Comments (deliberately written before open review)

1. Please attempt to make the manuscript more concise. It seems quite long for the material presented.
2. I am aware of, and have participated in the Nuffield STEM initiative that has been running for at least 10 years (https://www.nuffieldfoundation.org/contact). By linking school students to active research, placing them in universities it performs a similar role, and includes physics within its remit (i.e. one of my students investigated an element of geophysics, winning an award).
3. The paper is currently very UK-centric for an international journal.  Please make an attempt to identify and acknowledge other initiatives globally. It is difficult to believe that these do not exist, but if they do not, then please argue this case explicitly.
   - RISE programme at Stanford - summer internship programme. https://oso.stanford.edu/programs/disciplines/20-physics
   - https://www.sas.upenn.edu/summer/programs/high-school/experimental-physics
   - 'ANU extension' https://physics.anu.edu.au/engage/outreach/
4. If PRiSE is scalable, can you provide a simplified diagram that others could use to set up similar programmes, perhaps in other countries or other scientific fields? If it is purely physics (excluding physics in related disciplines) and only in the UK, state limitations on scope at the start. [see RC3 point 3]

L106 - "complete lack"? It would be good to see a review of other papers investigating schemes that use research in school as a method; these might be academic papers, but internal evaluations of these schemes or grey-literature they have published would be beneficial here.  Please add a paragraph.

L575-580 ..... discussion of causes of this expected later.

L593 ... how were the categories / themes defined? Add reference for analysis method please.

L806 - some formatting issues with references.

---

## Author Response (AR2)

**Response to reviewers**

**L4 - Please satisfy yourself that 'provision framework' is a thing. Sounds a little odd, but I can't immediately suggest a change.**

We have found several examples of this phrase used elsewhere, e.g.

- https://www.gov.uk/government/publications/review-your-remote-education-provision
- https://bidstats.uk/tenders/2020/W18/725876996
- https://www.dorsetnexus.org.uk/Article/56474

**L8 - comma before 'with'?**

This change has been made.

**L47 - Nice that international experience is recognised.**

**L50 - 'This review ....' ?**

This change has been made.

**L52 - This chimes with my experience of supervising high-school students on extended projects.**

**L55-65 - Good to place PRiSE in the context of other related schemes so that it's benefits can be clearly identified.**

**L67 - rather than '.... typically one ....' <5 perhaps might be fairer than 1, from my memory of Nuffield 2-3 was quite common although 1 was not unusual. '1 to 3'? It doesn't harm your point.**

We have changed this to 1-3 as based on the report of Paull and Xu (2017) for Nuffield Placements, which shows that while the majority of applicants (~55%) and placements (~75%) is just 1 per school, the average number per school is in the range 1-3 (skewed by the long tail).

**L135 - Use of supplementary material to avoid lengthy digressions from the stated point of the paper works.**

**L144-155 - Please consider placing this in a sub-section 2.1.1 highlighting ethics e.g. 'Ethical considerations'. This is something of importance to GC's ethos. If you choose to, renumber following sections i.e. 2.1.2 for Roles of Teachers etc ....**

We have made 'Ethical considerations' a dedicated subsubsection.

**L195 - Motivation of academics is a useful and important aspect to consider. Well put.**

**L198 - I agree it's definitely 'can' not 'is'. This is not as developed as it could be. Hillier et al (2019) 'Demystifying Academics' in GC (https://gc.copernicus.org/articles/2/1/2019/) includes an analysis of how impact is included in academics promotion criteria in the context of how they/we are motivated.**

We now reference this paper for interested readers.

**L557-559 - Your choice, but a somewhat negative tone. e.g. L559 instead of 'no data' you could just state that 'We have also gathered data on the impact of the PRiSE initiative, which is assessed elsewhere (Archer & Witt, 2020)'**

We have made this change.